# Contrast limited adaptive histogram equalization (CLAHE) and colour difference histogram (CDH) feature merging capsule network (CCFMCapsNet) for complex image recognition

Steve Okyere-Gyamfi[1,2]*, Michael Asante[1], Kwame Ofosuhene Peasah[1], Yaw Marfo Missah[1], Vivian Akoto-Adjepong[3]

1 Department of Computer Science, Kwame Nkrumah University of Science and Technology, Kumasi, Ghana, 2 Department of Computing and Information Sciences, Catholic University of Ghana, Sunyani, Ghana, 3 Department of Computer Science and Informatics, University of Energy and Natural Resources, Sunyani, Ghana

* steve.og@cug.edu.gh

## Abstract

To enhance crop yield, detecting leaf diseases has become a crucial research focus. Deep learning and computer vision excel in digital image processing. Various techniques grounded in deep learning have been utilized for detecting plant leaf diseases; however, achieving high accuracy remains a challenge. Basic convolutional neural networks (CNNs) in deep learning struggle with issues such as the abnormal orientation of images, rotation, and others, resulting in subpar performance. CNNs also need extensive data covering a wide range of variations to deliver strong performance. CapsNet is an innovative deep-learning architecture designed to address the limitations of CNNs. It performs well without needing a vast amount of data in various variations. CapsNets have their limitations, such as the encoder network considering every element in the image and the crowding issue. Due to this, they perform well on simple image recognition tasks but struggle with more complex images. To address these challenges, we introduced a new CapsNet model known as CCFM-CapsNet. This model incorporates CLAHE to reduce image noise and CDH to extract crucial features. Also, max-pooling and dropout layers are incorporated in the original CapsNet model for identifying and classifying diseases in apples, bananas, grapes, corn, mangoes, pepper, potatoes, rice, tomato and also for classifying fashion-MNIST and CIFAR-10 datasets. The proposed CCFM-CapsNet demonstrates significantly high validation accuracies, achieving 99.53%, 95.24%, 99.75%, 97.40%, 99.13%, 100%, 99.77%, 100%, 98.54%, 93.48%, and 82.34% with corresponding parameters in millions(M) 4.68M, 4.68M, 4.68M, 4.68M, 4.79M, 4.63M, 4.66M, 4.68M, 4.84M, 2.39M, and 4.84M for the datasets aforementioned respectively, outperforming the traditional CapsNet and other advanced CapsNet models. Consequently, the CCFM-CapsNet model can be utilized effectively as a smart tool for identifying plant diseases and

**Data availability statement:** Apple, Grape, Corn, Pepper, Potato, and Tomato datasets: https://github.com/spMohanty/PlantVillage-Dataset. Banana: https://data.mendeley.com/datasets/9tb7k297ff/1. Mango: https://data.mendeley.com/datasets/hxsnvwty3r/1. Rice: https://data.mendeley.com/datasets/fwcj7st-b8r/1. Fashion-MNIST: https://www.kaggle.com/datasets/zalando-research/fashionmnist. CIFAR-10: https://www.cs.toronto.edu/~kriz/cifar.html.

**Funding:** The author(s) received no specific funding for this work.

**Competing interests:** The authors have declared that no competing interests exist.

also in achieving Sustainable Development Goal 2 (Zero Hunger), which aims to end global hunger by the year 2030.

---

## 1. Introduction

The primary aim of smart farming is to create innovative solutions that ensure the long-term sustainability of humanity. One of the main obstacles to food security is protecting crops from various biological and non-biological threats, with plant diseases being a significant challenge. These diseases not only devastate crops or reduce their quality but also lead to the use of pesticides that contaminate the soil, eventually making it unfit for future cultivation. Most plants are grown in developing countries, where most farmers lack awareness of the diseases that can impact these crops and the methods to control them. This situation negatively impacts crop production, potentially leading to hunger and a decrease in the essential nutrients needed for good health. Therefore, it is crucial to identify diseases promptly and precisely to avoid agricultural losses. However, most identification techniques are manual, making them time-consuming, labour-Intensive, and prone to errors, particularly in the early stages.

A significant advancement was made possible by the introduction of deep learning (DL), particularly in the sector of agriculture. DL techniques, namely Artificial Neural Networks (ANN) and CNN, have emerged as the frequent widely utilized techniques for the identification and diagnosis of diseases in plants diseases imaging technology [1]. Nonetheless, each of these architectures has its advantages and disadvantages that often impact the model's performance. Both architectures need large datasets to train models effectively, and as image resolution increases, more trainable parameters are added, which affects the model's computing cost and runtime. Noord and Postma claim that ANNS lack flexibility and make it difficult to customize the model. Furthermore, ANN also has problems with gradients that explode and shrink [2]. Recent advances in image classification (especially in crop disease detection) have been dominated by CNNs such as EfficientNet and ResNet, DenseNet, as well as transformer-based models like the Vision Transformer (ViT) [3–8]. These models have also demonstrated impressive performance across large-scale benchmark datasets such as CIFAR-10, fashion-MNIST, ImageNet, etc., often surpassing human-level accuracy [9]. Their success is largely attributable to hierarchical feature extraction (in CNNs) and global self-attention mechanisms (in ViTs), both of which enable strong generalization when large volumes of labeled data are available. For example, EfficientNet employs a joint scaling strategy to proportionally scale the width, depth, and resolution, yielding top-tier accuracy [5]. Similarly, ViT leverages patch embeddings and multi-head attention to capture long-range dependencies, pushing the boundaries of image classification performance [6].

While these models are highly influential in advancing the field, they also highlight the gaps that Capsule Networks (CapsNets) are designed to address. CNNs and

ViTs often struggle with preserving spatial hierarchies and viewpoint equivariance, which can be critical in fine-grained recognition tasks or cases with limited data, such as those in health, agriculture, etc. [10,11].

As CNNs go deeper, they need broad data coverage to prevent overfitting [12]. They also maintain good performance as the model depth increases; however, this introduces challenges, including significant parameter numbers, amplified complexity, increased memory requirements, and intensive computing requirements. To ensure CNNs and ViTs perform effectively and generalize well to new data, time-consuming and labour-intensive data augmentation techniques are necessary [12].

CapsNets, through their routing-by-agreement mechanism, are inherently designed to encode part–whole relationships, making them more interpretable and potentially more robust in low-data regimes addressing the limitations of CNN's [11].

Capsules need less training data, are less impacted by class imbalance, and are more resilient to changes in spatial orientation. Despite these benefits of CapsNets, they also have some disadvantages [1]. Due to the "crowding" problem, their performance is poor on complex images that have diverse backgrounds [12]. This issue affects various datasets such as CIFAR-100, CIFAR-10, wild plant images, medical images, multilabel images, and many more. The Capsule network's sensitivity to the image background makes it prone to misclassification.

Additionally, CapsNet attempts to account for every detail in the image. Due to these properties, the network's performance can decline when processing detailed infected images. The encoder network's ineffective ability to extract features considerably hinders CapsNet's performance [13,14]. Consequently, enhancing the current capsule network algorithm is necessary to effectively classify these images.

To enhance the categorization of plant ailments employing CapsNet, we incorporated CapsNet with dynamic routing [11]. Additionally, we introduced CDH in conjunction with CapsNet to extract crucial features for the first time. Moreover, we integrated CLAHE to improve visual clarity by reducing inherent noise, thereby enhancing the feature extraction capabilities of the encoder network.

It is worth highlighting that the current investigation is devoted to the advancement of CapsNet architectures rather than proposing a direct replacement for CNNs or ViTs. The comparisons in this study are therefore made against established CapsNet variants to ensure that the proposed contributions are evaluated fairly within the same model family. Nonetheless, discussing CNN- and transformer-based models remains relevant because they are part of the cutting-edge techniques in image classification. Their strengths and limitations provide a useful backdrop for understanding why improving CapsNets is significant. In particular, CNNs and ViTs, together with their variants are known to require extensive data augmentation and large-scale training datasets to generalize well [6–8], whereas CapsNets offer a principled approach to capturing spatial hierarchies with potentially fewer samples [10,11]. Thus, by improving CapsNet performance, our work contributes toward bridging the gap between biologically inspired models and the efficiency of mainstream architectures.

While CNN-based models and ViTs set the benchmark for large-scale classification performance, the relevance of CapsNets lies in their unique ability to capture pose and spatial relationships in a way that conventional models cannot and its ability to perform well on smaller datasets. The proposed CCFM-CapsNet therefore extends the applicability of CapsNet-based models, demonstrating that with appropriate architectural refinements, CapsNets can achieve competitive or superior performance suitable for resource-constrained devices, particularly in domains like agriculture and health, where interpretability, robustness, and data efficiency are critical.

To elaborate further, our work's contributions can be outlined as follows:

• A novel architecture named CCFM-CapsNet, based on CapsNet, is introduced for the classification of plant diseases.

• CDH is used as a feature extraction component alongside CapsNet, and we examined how it influences the models' effectiveness.

• A comparative assessment was performed between the CCFM-CapsNet and alternative cutting-edge CapsNet models.

 

- A thorough visualization of layer outputs is performed to facilitate explainable artificial intelligence (XAI).

- The proposed CCFM-CapsNet achieves efficiency by keeping parameters minimal, which cuts down on memory and processing costs, enhancing adaptability to less resource-intensive devices.

This article is structured into several sections: Section 2 examines prior research, Section 3 elaborates on the methods adopted, Section 4 showcases the experimental results alongside discussion, and Section 5 draws conclusions and suggests possible future directions.

## 2. Related works

Problems associated with manual plant disease detection have prompted the need for automated and intelligent models. There has been extensive CNN domain research, but relatively little CapsNet domain research. Within the field of CapsNet-based plant disease recognition, Zhang and colleagues suggested a model for disease detection by modifying the convolution kernel in the residual network (ResNet) to a 3x3 kernel and adding an attention mechanism to capture relevant properties. The enhanced ResNet was integrated into CapsNet. The SE-SK-CapResNet which is an aggregated model, achieved validation accuracies of 98.58% on PlantVillage, 95.08% on AI Challenger 2018, and 97.19% on Tomato disease datasets [15]. Also, Abouelmagd and co-authors presented a method for computer vision. During training, they employed data augmentation and pre-processing strategies to prevent overfitting. This method gained a validation accuracy of 96.39% on tomato dataset that comprises ten classes [16]. Mensah and his co-author developed a dual-input CapsNet model that sends input through two separate convolution layers before combining their outputs, which are then sent to the standard CapsNet layers. This architecture was evaluated on the tomato and CIFAR-10 datasets, attaining 93.03% and 76.58% accuracies and, 6.04 M and 5.48 million (M) parameters generated, respectively [17]. Peker presented a new CapsNet method using a multi-channel ensemble CapsNet. The innovation is using several CapsNets, each trained independently on images that have undergone different pre-processing methods. By combining these networks, the model improves the detection of plant diseases by taking advantage of diverse data features. This architecture was assessed on a tomato dataset containing 10 classes, achieving an accuracy of 98.15% [18]. Mensah and his team again suggested two new versions of CapsNet: Multi-Input CapsNet and Shallow CapsNet. The Shallow CapsNet variant included a normalizer, a modified squashing function, and an LBP layer integrated into the traditional CapsNet structure. On tomato, fashion-MNIST, and CIFAR-10, accuracies of 97.33%, 92.70%, and 75.75%, and 4.1 million (M), 2.5 M, 4.6M generated parameters were attained. The multi-input CapsNet combined aggregated features extracted from 3 convolution layers and fed it into the conventional CapsNet model. On the same dataset, 94.04%, 91.45%, and 63.95% validation accuracies were attained by the model with 4.0M, 2.2M, and 4.3M generated parameters, respectively [19]. With the use of Gabor filters for encoding spatial and texture arrangements and max pooling to reduce vector dimensions and gain only prominent characteristics, Mensah and the team once again suggested a CapsNet architecture to identify tomato disease employing the PlantVillage dataset, which gained a validation accuracy of 97.98% and generated 8,708,128 parameters [20]. Also, Oladejo and Ademola suggested an optimized CapsNet architecture by changing the fully connected layer's neuron number from three to two (512 and 1024) in the traditional CapsNet to decrease time for training and increased the processing speed of the network by incorporating a momentum optimizer. Their model gained a better validation accuracy of 95% on banana leaf disease detection [21]. Altan suggested a CapsNet model by changing the neuron numbers in the fully connected layer to 960, 768, and 4096 for the first, second, and third layers of the conventional CapsNet. The model performed very well on bell pepper disease classification and gained 96.37%, 97.49%, and 95.76% sensitivity, specificity, and accuracy, respectively [22]. Verma et al employed CapsNet for the recognition of potato diseases, using the PlantVillage potato dataset. Results show the model achieved 91.83% and generated 9,856,768 parameters [23]. Mensah and his team suggested a CapsNet model that integrates Local Binary Pattern and K-Means routing. Their routing algorithm substitutes the dynamic routing algorithm's SoftMax, dot product, squashing function, and normalizer with the sigmoid function, squared Euclidean distance, and simple squash. Results on citrus, tomato,

 

and maize datasets (plant village) attained validation accuracies of 98.21%, 98.80% (with 5.12 million parameters),97.99%, respectively [24]. Xu and the team also suggested a CapsNet model by combining two (2) inception modules with different rates of dilations to boost receptive fields of convolution to gain multi-layered characteristics from diseased apple leaf images. On apple diseases, the architecture gained 93.16% accuracy [25]. By substituting sigmoid with SoftMax, k-means clustering for dynamic routing, and CNN with Local Binary Pattern CapsNet, Mensah and team suggested an architecture. The model attained validation accuracies of 96.79%, 99.41%, 98.06%, 92.72%, 75.80%, and 99.68%, for maize, citrus, tomato, fashion-MNIST, and MNIST dataset. The model generated 8.4 M parameters for the maize, tomato, and citrus, 5.2 M for CIFAR-10, and 2.8 M for the MNIST and fashion-MNIST datasets [26]. Vasudevan and Karthick suggested a hybrid architecture for classifying diseases in grapes. They employed a graph-based technique to determine the area of the leaf and subsequently utilized a Generative Adversarial Network to augment the dataset. This model was computationally expensive. The disease classification was finally done using CapsNet and achieved 97.63% accuracy on the dataset from PlantVillage and other captured ones [27]. Mensah and his friends recommended employing CapsNet and Gabor to detect abnormalities in images of citrus and tomato diseases that are disfigured and obscured, using the Plant Village dataset. Based on their experimental findings, the model attained 98.13% validation accuracy using 12M parameters for detecting tomato diseases and 93.33% accuracy in identifying citrus diseases [28]. Verma and team introduced an enhanced method for computing features using Squeeze and Excitation (SE) Networks, which precede the original Capsule Networks (CapsNet) in the classification pipeline to assess plant disease extent. AleNet and ResNet was each added to CapsNet to produce two SE networks. Focusing on Late Blight disease in tomatoes, they utilized the PlantVillage dataset and classified the images into healthy, early, middle, and late stages. The SE-Alex-CapsNet model attained 92.1% accuracy with 9 million parameters, while the SE-Res-CapsNet model attained a higher accuracy of 93.75% with 19 million parameters [29]. Andrushia and his team presented an architecture to recognize diseases in grapes utilizing convolutional capsule networks. Their innovative approach involves introducing convolutional layers to precede the primary capsule layer, to help lower capsule numbers and accelerate the process of dynamic routing. This method was evaluated on both augmented and non-augmented datasets and attained a 99.12% accuracy in identifying grape leaf diseases from the Plant Village dataset [30]. Anant unveiled a novel deep learning technique named AppleCaps, specifically designed to accurately classify various disease types, overcoming the spatial invariance problem found in CNNs. When evaluated on an augmented dataset, the AppleCaps model achieved an 87.06% accuracy in detecting diseases on apple leaves [31].

The discussed CapsNet models have demonstrated good performance on various datasets. Meanwhile, to be able to efficiently and effectively detect plant diseases, more efficient and low-parameterised models that can be deployed on resource-constrained devices are needed.

## 3. Methods

### 3.1. Original capsule network

A capsule is a neuron collection with each connected to an activity vector describing distinct instantiation attributes used to identify a particular object or a part of it [11]. The direction and magnitude of the vector indicate the object's probability and estimated pose. These vectors are transmitted from lower-level capsules to higher-level ones, with coupling coefficients governing the connections between the layers. When a lower-level capsule's prediction closely matches what a higher-level capsule produces, they become more strongly linked. This strength of connection is quantified using the SoftMax function. Routing by agreement happens when a capsule spot a close-knit collection of prior predictions, which signals a high chance of the object being present.

The process begins with the computation of the prediction vector, as shown in Equation 1;

$$\hat{u}_{j|i} = W_{ij}u_i \tag{1}$$

The mapping from a capsule in the upper block, denoted as $\hat{u}_{j|i}$ for the $j^{th}$ capsule based on input from the $i^{th}$ lower-level capsule, is determined by multiplying the lower-level capsule's output vector $u_i$ with a weight matrix $W_{ij}$. Building upon this, as described in Equation 2, the coupling coefficients which indicate the agreement between capsules in adjacent layers are computed using the SoftMax function.

$$C_{ij} = \frac{\exp(b_{ij})}{\sum_k \exp(b_{ik})}$$

(2)

Here, $b_{ij}$ represents the log probability among two (2) capsules, which is initially set to zero, and $k$ denotes the number of capsules. The $j^{th}$ layer input vector $s_j$, representing a weighted total of vectors for $\hat{u}_{j|i}$ determined by the routing algorithm, is computed as seen in Equation 3;

In this setup, $b_{ij}$ starts at zero and signifies the logarithm of the probability connecting two capsules. Here, $k$ is the total count of capsules. The input vector $s_j$ for the $j^{th}$ layer is calculated using Equation 3. This vector $s_j$ is essentially a weighted sum of the prediction vectors $\hat{u}_{j|i}$, where the weights are determined dynamically by the routing algorithm.

$$s_j = \sum_{i=1}^{N} c_{ij} . \hat{u}_{j|i}$$

(3)

Finally, to obtain a likelihood value ranging 0–1, a special function (Equation 4) is applied. This function combines unit scaling with a squashing mechanism to ensure the output stays within the desired range.

$$v_j = \frac{\|s_j\|^2}{1 + \|s_j\|^2} \frac{s_j}{\|s_j\|}$$

(4)

Equation 5 provides the means to quantify the discrepancy between the predicted and actual outputs of the capsules situated in the network's final layer.

$$L_k = T_k \, max(0, m^+ - \|v_k\|)^2 + \lambda(1 - T_k) \, max(0, \, \|v_k\| - m^-)^2$$

(5)

$T_k = 1$ when the $k$ class is activated, and 0 when deactivated. The hyperparameters $m^+$, $m^-$, and $\lambda$ are estimated at the learning phase.

Whether class $k$ is activated ($T_k = 1$) or not ($T_k = 0$) depends on learned hyperparameters $\lambda$, $m^-$, and $m^+$.

### 3.2. Colour difference histogram (CDH)

The CDH technique focuses on edge and colour orientation, as well as perceptually consistent colour differences, encoding these attributes in a way that mimics the human vision system. It serves as an innovative visual attribute descriptor that integrates colour, edge orientation, and perceptually consistent colour differences, while also considering spatial layout.

**3.2.1. CDH feature map detection.** To begin with, CDH performs colour and edge quantization. The next step deals with identifying microstructures by applying friction to a single pixel and using a 3x3 filter for both quantization [32]. The central value of the filter is evaluated against its surrounding eight values. This process generates edge maps and colour maps. Using these maps, we can compute the difference $\Delta$) in edge orientation and colour intensity for each $L*a*b*$ component, storing them as colour and edge features. Finally, these two features are integrated into a composite histogram. The quantized image values at the position $C(x, y)$ are represented as $w \in 0, 1, \ldots, W - 1$. Let $(x, y)$ and $(x'y')$ denote neighbouring pixel locations with their corresponding colour indices $C(x, y) = w_1$ and $C(x'y') = w_2$. The edge orientation image $\theta(x, y)$ values are represented as $v \in 0, 1, \ldots, V - 1$, with angles at $(x, y)$ and $(x'y')$ being $\theta(x, y) = v_1$ and

$\theta(x'y') = v_2$ respectively. For nearby pixels at a distance $D$, and given the quantization levels $W$ for colour and $V$ for edge orientations, CDH can be defined using Equations 6 and 7.

$$H_{color}(C(x,y)) = \begin{cases} \sum \sqrt{\left(\Delta L^2\right) + \left(\Delta a^2\right) + \left(\Delta b^2\right)} \\ where\ \theta(x,y) = \theta(x'y')\,;\,max\,(|x-x'|)\,,\,(|y-y'| = D) \end{cases} \tag{6}$$

$$H_{ori}(\theta(x,y)) = \begin{cases} \sum \sqrt{\left(\Delta L^2\right) + \left(\Delta a^2\right) + \left(\Delta b^2\right)} \\ where\ C(x,y) = C(x'y')\,;\,max\,(|x-x'|)\,,\,(|y-y'| = D) \end{cases} \tag{7}$$

Where ΔL, Δa, and Δb represent variation among 2 colour pixels, with $D = 1$ being the standard. Equation 8 expresses the CDH feature, where $W$ designates the edge orientation and $V$ represents the colour quantization.

$$H_{CDH} = H_{color}(0),\ H_{color}(1)\dots H_{color}(W-1),\ H_{ori}(0), H_{ori}(1)\dots H_{ori}(V-1) \tag{8}$$

where the colour histogram is represented by $H_{color}$, and the edge orientation histogram is represented by $H_{ori}$. To demonstrate, if we designate the dimensionality of colour quantization as 72 and edge orientation as 18, then the merged features for image retrieval equal 90 dimensions. These combined features are denoted as H.

### 3.3. Contrast limited adaptive histogram equalization (CLAHE)

The CLAHE method [33] is applied as a pre-processing step not only to reduce noise in images but also to effectively boost overall contrast. This contrast enhancement is particularly advantageous when the subject and background exhibit comparable contrast. Histogram equalization aims to create a brighter mammogram with better contrast by distributing pixel intensities more uniformly. CLAHE preprocessing component also improves generalization by standardizing local contrast and reducing overfitting due to lighting variability.

To improve contrast in single-channel mammography images, models use CLAHE. Instead of processing the whole image, CLAHE works on smaller sections called tiles. Contrast enhancement is applied to each small section of the image independently in CLAHE. This process aims to make the brightness levels within each section more evenly distributed. By working on these smaller areas, CLAHE is good at reducing noise and ultimately produces better image quality. Images are prepared using the CLAHE method as shown in Equation 9.

$$CLAHE_{pre-processing} = \frac{Processed_{image}}{Original_{image}} \tag{9}$$

Where the contrast values of the raw image and the noise-reduced image are represented by $Original_{image}$ and $Processed_{image}$, respectively. Equation 10, the contrast equation, then utilizes these values to eliminate noise and improve the image.

$$Contrast = \frac{Gray\_Level_{value\ of\ image}}{Background_{value\ of\ image}} \tag{10}$$

### 3.4. Proposed CCFM-CapsNet model

The CCFM-CapsNet model proposed utilizes CDH and CLAHE techniques to extract crucial features, amalgamating edge orientation, colour, and perceptually consistent colour differences while considering spatial layout to enhance textural

features in input images. The decision to employ these strategies stems from the superior performance of numerous model alterations, aiding in noise reduction and feature extraction to minimize misclassification. CLAHE is employed to boost image contrast and restrict the neighbouring pixel procedure to diminish noise, improve image quality, and mitigate overfitting. Conversely, CDH enables the extraction of vital features by combining colour and edge orientations.

Fig 1 presents the suggested CCFM-CapsNet framework, which contains a CDH, CLAHE, four 3x3 layers of convolutions, four layers of max-pooling for feature extraction, a layer of dropout to randomly deactivate neurons during training to help reduce the model's reliance on individual neurons to improve robustness, promote generalization on unseen data, and help prevent overfitting, a primary capsule, a class capsule, and 3 layers that are fully connected layers. The CCFM-CapsNet is made up of two lanes receiving input from the input image. Features from the input image which is of dimension 32x32x3 are subjected to Lane1(L1) through a CLAHE layer without generating additional parameters. The CLAHE layer 32x32x3 feature map passes through the Conv1 layer (256 filters), with kernel size of 3x3, and a 1 stride, which produces a 30x30x256 feature map. This is followed by a batch normalizer. This resulting map is forwarded to a max-pooling layer (MP1) to give a 15x15x256 feature map. The output of the max-pooling layer (MP1) is forwarded to another convolution layer Conv2 with 128 filters, 3x3 kernel size, and stride of 1, which produces a 13x13x128 feature map and then batch normalization is performed. This resulting map is forwarded to a max-pooling layer (MP2) to give a 7x7x128 feature map. Features from the input image are also subjected to Lane1(L2) through a CDH layer which does not generate additional parameters. Results from the CDH layer (32x32x3 feature map) are sent to the Conv3 layer (256 filters), with a step-size of 1 and a 3x3 kernel size, producing a feature-representation of 30x30x256 followed by

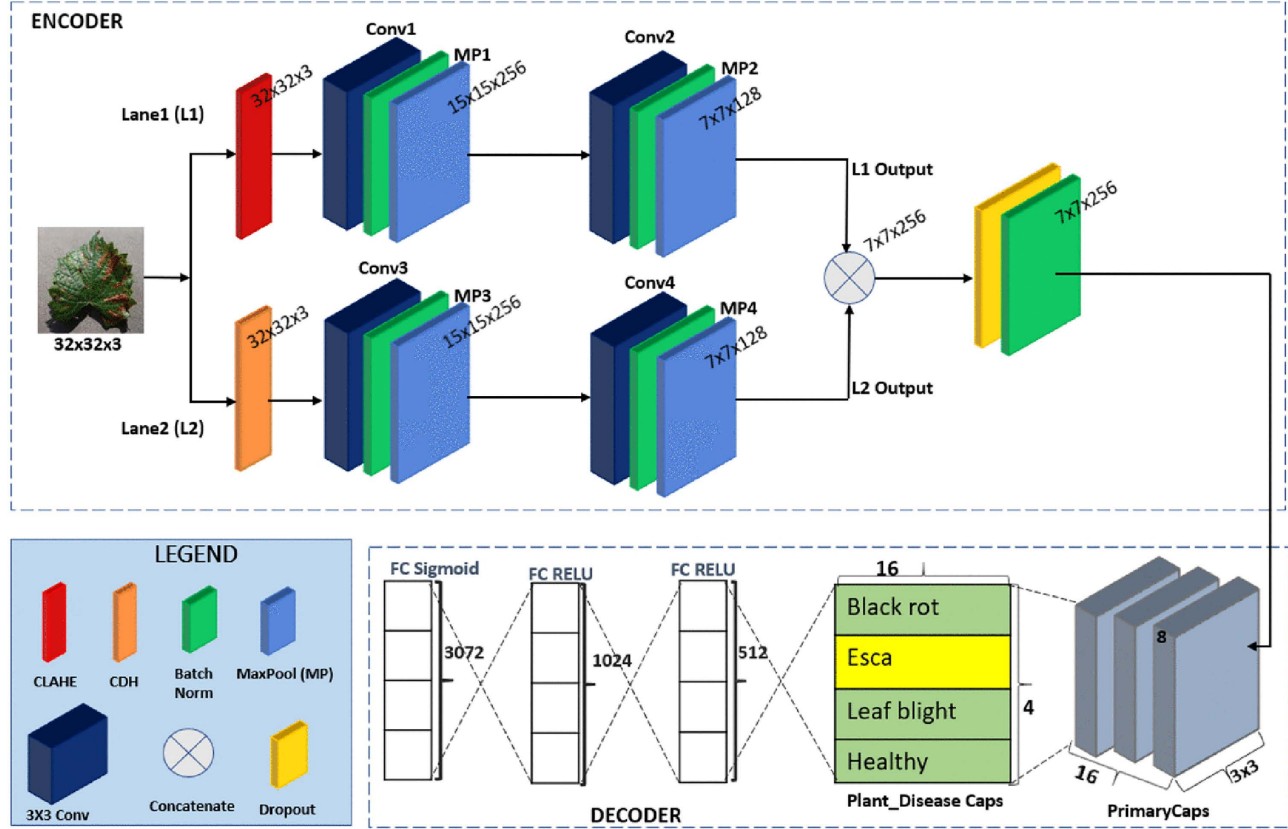

**Fig 1. Proposed CCFM-CapsNet model architecture.**

 

batch normalization. This resulting map is forwarded to a max-pooling layer (MP3) to give a 15x15x256 feature map. The results from the MP3 are forwarded to another convolution layer Conv4 (128 filters), kernel size of 3x3, and a stride of 1, which produces a 13x13x128 feature map, then a batch normalization is applied. This resulting map is forwarded to a max-pooling layer (MP4) to give a 7x7x128 feature map. Outputs from L1 and L2 are merged, producing a 7x7x256 map, allowing the merging of features to help in the learning of complex and integrated features by the model. The concatenated feature map is then sent through a dropout layer with a batch normalizer, then through the Primary Capsule having 16 channels with 8 dimensions instead of 32 dimensions to help lower the parameter count, computational cost, and also mitigate overfitting, a kernel of size 3x3 and a stride of 2.

The Plant_DiseaseCaps (class capsule) made up of 16D capsules which accounts for all the distinct classes within the dataset, receives its input from the primary capsule and sends its output for reconstruction. The decoder layer then receives the output and decodes the entity properties. The decoder used in this model is composed of three neurons that are fully connected, comprising 512, 1024, and 3072 respectively. The CDH, CLAHE, convolution and max pooling layers help to extricate dominant features from the subjected images for better extraction of features and classification. Grape leaf diseases were used for the illustration in Fig 1.

### 3.5. Datasets

Eleven independent datasets (i.e., nine plant disease datasets from Kaggle, Mendley, etc, and two benchmark datasets) were used independently to run and assess the original and CCFM-CapsNets effectiveness, and compared with various cutting-edge CapsNet models found in the literature that used the same datasets to evaluate their performance. This includes the Apple, Corn, Grape, Pepper, Tomato, Potato, Banana, Mango, Rice, Fashion-MNIST, and CIFAR-10 datasets.

The datasets of **Corn**, **Apple**, **Pepper**, **Tomato, Grape,** and **Potato** were sized 256x256, and are included in the Plant Village dataset [34].

**Banana:** Consists of varying dimensions ranging from *2230x4000 to 3120x4208* [35].

**Mango:** Consists of varying dimensions ranging from 240x240 to 320x240 [36].

**Rice:** Consists of varying dimensions ranging from 216x289 to 344x516 [37].

**Fashion-MNIST:** Consists of grayscale images, of size 28x28, and is complex [38].

**CIFAR-10:** consists of 32x32x3. They have different backgrounds and are more complex than Fashion-MNIST [39].

Table 1 presents the total image count for each of the 11 datasets, along with the number of classes and the sample size per class.

Most of the datasets for plant disease are significantly imbalanced, with images that are very similar to each other and backgrounds that are not uniform. The pre-processing step utilized to these datasets was rescaling the images to a dimension of 32 x 32 x 3 to maintain manageability, except for the Fashion-MNIST dataset. Additionally, an 80:20 split technique was employed.

### 3.6. Experimental setup

**3.6.1. Hyperparameter tuning.** The following hyperparameters were chosen for performance evaluation of both the original and CCFM-CapsNetss training and testing, and compared with various cutting-edge CapsNet models found in the literature that used the same datasets to assess their performance:

- Learning rate: 0.001 with exponential decay (decay rate = 0.9)

- Batch size: 100

- Epochs: 200

- Routing iterations: 3

**Table 1. Summary of the 11 independent datasets used separately for model training and evaluation.**

| S/N | Dataset | Total no. of images | No. of Classes | Classes and no. of samples | Number of samples (images) |
|---|---|---|---|---|---|
| 1 | Apple | 3,171 | 4 | 0: Apple_scab | 630 |
| | | | | 1: Black_rot | 621 |
| | | | | 2: Cedar_apple_rust | 275 |
| | | | | 3: healthy | 1645 |
| 2 | Grape | 4,062 | 4 | 0: Black_rot | 1180 |
| | | | | 1: Esca_(Black_Measles) | 1383 |
| | | | | 2: Leaf_blight_(Isariopsis_Leaf_Spot) | 1076 |
| | | | | 3: healthy | 423 |
| 3 | Corn | 3,852 | 4 | 0: Cercospora_leaf_spot, Gray_leaf_spot | 513 |
| | | | | 1: Common_rust | 1192 |
| | | | | 2: Northern_Leaf_Blight | 985 |
| | | | | 3: healthy | 1162 |
| 4 | Pepper | 2,475 | 2 | 0: healthy | 1478 |
| | | | | 1: Bacterial_spot | 997 |
| 5 | Potato | 2,152 | 3 | 0: Healthy | 152 |
| | | | | 1: Early_blight | 1000 |
| | | | | 2: Late_blight | 1000 |
| 6 | Tomato | 18,160 | 10 | 0: Bacterial_spot | 2127 |
| | | | | 1: Early_blight | 1000 |
| | | | | 2: Late_blight | 1909 |
| | | | | 3: Leaf_Mold | 952 |
| | | | | 4: Septoria_leaf_spot | 1771 |
| | | | | 5: Spider_mites Two-spotted_spider_mite | 1676 |
| | | | | 6: Target_Spot | 1404 |
| | | | | 7: Tomato_mosaic_virus | 373 |
| | | | | 8: Tomato_Yellow_Leaf_Curl_Virus | 5357 |
| | | | | 9: Healthy | 1591 |
| 7 | Banana | 937 | 4 | 0: Cordana | 162 |
| | | | | 1: Pestalotiopsis | 173 |
| | | | | 2: Sigatoka | 473 |
| | | | | 3: Healthy | 129 |
| 8 | Mango | 4,000 | 8 | 0: Anthracnose | 500 |
| | | | | 1: Bacterial Canker | 500 |
| | | | | 2: Cutting Weevil | 500 |
| | | | | 3: Die Back | 500 |
| | | | | 4: Gall Midge | 500 |
| | | | | 5: Powdery Mildew | 500 |
| | | | | 6: Powdery Mildew | 500 |
| | | | | 7: Healthy | 500 |
| 9 | Rice | 5,932 | 4 | 0: Bacterial_blight | 1584 |
| | | | | 1: Blast | 1440 |
| | | | | 2: Brown_spot | 1600 |
| | | | | 3: Tungro | 1308 |

*(Continued)*

**Table 1.** (Continued)

| S/N | Dataset | Total no. of images | No. of Classes | Classes and no. of samples | Number of samples (images) |
|---|---|---|---|---|---|
| 10 | Fashion-MNIST | 70,000 | 10 | 0: T-shirt/top | 7000 |
| | | | | 1: Trouser | 7000 |
| | | | | 2: Pullover | 7000 |
| | | | | 3: Dress | 7000 |
| | | | | 4: Coat | 7000 |
| | | | | 5: Sandal | 7000 |
| | | | | 6: Shirt | 7000 |
| | | | | 7: Sneaker | 7000 |
| | | | | 8: Bag | 7000 |
| | | | | 9: Ankle boot | 7000 |
| 11 | CIFAR-10 | 60,000 | 10 | 0: airplane | 6000 |
| | | | | 1: automobile | 6000 |
| | | | | 2: bird | 6000 |
| | | | | 3: cat | 6000 |
| | | | | 4: deer | 6000 |
| | | | | 5: dog | 6000 |
| | | | | 6: frog | 6000 |
| | | | | 7: horse | 6000 |
| | | | | 8: ship | 6000 |
| | | | | 9: truck | 6000 |

- Reconstruction loss coefficient ($\lambda$): 0.392

- All images in the various datasets were resized to 32X32X3, except Fashion-MNIST, which was 28x28x1

- For each of the 11 independent datasets used in this study, we applied an 80:20 train-test split. All splits were stratified to maintain class distribution.

### 3.6.2. Experimental pipeline.

- All model training and testing took place in Keras, running on top of TensorFlow, using a 64-bit Windows computer equipped with an NVIDIA GeForce RTX 2080 SUPER GPU (8 GB).

- Training was performed separately for each of the 11 datasets using the same pipeline to ensure fair comparisons.

- The top-performing model was saved and used for evaluation.

- The original CapsNet implementation from Xifeng Guo from Sabour was used as the base and modified to accommodate the proposed architectural enhancements.

- Evaluation metrics included precision, accuracy, sensitivity or recall, F1-score, specificity, ROC-AUC, and PR-AUC.

## 3.7. Performance evaluation measures

The study employed a variety of metrics to assess classification performance:

- **Confusion matrix:** Offers a comprehensive summary of both accurate and inaccurate predictions, enabling the computation of metrics such as sensitivity (recall), precision, specificity, F1-Score, and accuracy from TP-true positive, FP-false positive, TN-true negative, FN-false negative.

- **Validation accuracy:** indicates the ratio of samples correctly classified compared to the overall number of instances. The overall validation accuracy reported reflects outcomes across all experiments, as shown in Equation 11.

$$Accuracy = \frac{TP + TN}{TP + FP + TN + FN} \tag{11}$$

- **Loss:** Measures how far the model's predictions deviate from the true labels. Margin loss was specifically used during assessment.

- **Precision (P):** Measure of how many of the predicted positives are truly positives. This is shown in Equation 12.

$$Precision(P) = \frac{TP}{TP + FP} \tag{12}$$

- **Recall (R)/ Sensitivity:** Measures how many of the actual positive instances are correctly detected by the model. This is shown in Equation 13.

$$\frac{Recall(R)}{Sensitivity} = \frac{TP}{TP + FN} \tag{13}$$

- **Specificity:** The proportion of true negatives identified among all actual negatives. This is shown in Equation 14.

$$Specificity = \frac{TN}{TN + FP} \tag{14}$$

- **F1-Score:** Combines precision and recall through their harmonic mean, ensuring a trade-off between them. This is shown in Equation 15.

$$F1 - Score = 2\left(\frac{P * R}{P + R}\right) \tag{15}$$

- **Area under the Curve (AUC):** Performance evaluation of models are assessed with Receiver Operating Characteristic (ROC) and Precision–Recall (PR) curves, which are especially informative for imbalanced datasets. Larger AUC values signify a strong discriminative capability.

- **Clustering analysis:** To gain insight into the feature distribution, t-SNE was used to visualize and interpret class capsule clusters generated by the model.

- **Model complexity:** The study also measured performance by analyzing computational efficiency, expressed through parameter count and memory usage.

## 4. Experimental results and discussion

Here, we showcase the outcomes of training the suggested architecture with images of plant leaves, CIFAR-10, and fashion-MNIST. We provide visual representations such as training loss and validation curves, AUC and confusion

matrices. The confusion matrices display actual classes in non-white columns and predicted output classes in non-white rows. Additionally, we conduct an ablation study to demonstrate the model's resilience and adaptability. Furthermore, we offer visualizations of class capsule clusters, activation maps, and image reconstruction, shedding light on the model's inner mechanisms and contributing to the field of XAI [40–42].

## 4.1. Performance assessment

For the CIFAR-10 dataset in Fig 2b, looking at epoch 20, a clear difference emerges. The traditional CapsNet, as described by Sabour and colleagues in 2017 [11], shows a concerning pattern: its validation accuracy for CIFAR-10 is not only low but starts to drop, implying that its performance on unseen data is getting worse. In contrast, your CCFM-CapsNet model behaves quite differently. Starting around epoch 15, its validation accuracy, which is already high, levels off and stays consistent until the end of the training. This stable, high accuracy suggests that your model is more reliable and generalizes better to new, unseen images from the CIFAR-10 dataset. Furthermore, the CCFM-CapsNet model demonstrated significantly higher accuracy across all tested datasets, including individual datasets for apples, bananas, grapes, maize, mangoes, pepper, potatoes, rice, tomatoes, CIFAR-10, and Fashion-MNIST, achieving scores of 99.53%, 95.24%, 99.75%, 97.40%, 99.13%, 100%, 99.77%, 100%, 98.54%, 82.34%, and 93.48% respectively as seen in Fig 2a and Fig 2b. Furthermore, this improved performance was achieved more quickly compared to the original model, which only reached accuracies of 92.91%, 80.95%, 93.49%, 92.59%, 78.38%, 59.68%, 95.36%, 99.24%, 88.59%, 63.58%, and 90.98% on the same datasets as seen in Fig 2a and Fig 2b. This implies that the CCFM-CapsNet is more robust and adaptable to new situations than the traditional CapsNet.

When working with datasets that are not balanced, like the tomatoes, apples, corn, grapes, and pepper datasets, just considering the test accuracy can be misleading and might not tell how well a model is actually performing on each individual category in a dataset. Hence, it is more equitable to deduce and assess the TP, FP, TN, FN, precision (PRE), accuracy (ACCU), sensitivity (SENS), F1-Score (F1) and specificity (SPEC), for both the CCFM-CapsNet and traditional CapsNet from the confusion matrices. The confusion matrices for mango and grape found in Fig 3 show that the CCFM-CapsNet model had fewer misclassifications than the traditional model. Tables 2 and 3 comprehensively summarize the performance assessment on the mango and grape datasets deduced from the confusion matrices. Upon reviewing Tables 2 and 3, and Figs 4 and 5, it becomes evident that the CCFM-CapsNet model outperforms the traditional one across all metrics, suggesting minimal misclassification and better generalization by the CCFM-CapsNet model.

Again, considering the ROC and PR curves of the suggested CCFM-CapsNet and traditional model using the dataset of mango (Fig 6) and CIFAR-10 (Fig 7). The CCFM-CapsNet method attained better values when assessed on the Mango dataset. The CCFM-CapsNet markedly surpassed the original CapsNet, achieving near-perfect ROC and PR scores of 100% and 99.75%, respectively, compared to the latter's 94.13% and 78.75%. Similarly, on the CIFAR-10 dataset, the CCFM-CapsNet showed substantial gains, reaching ROC and PR values of 96.20% and 88.50%, while the original CapsNet only attained 90.80% and 67.80%. The ROC curves visually confirm the CCFM-CapsNet's superior ability to differentiate between classes across various decision thresholds. Furthermore, its consistently high PR values across all datasets highlight its robustness in handling imbalanced data.

## 4.2. Ablation study

Conducting an ablation study allows insight into how each segment of a model influences performance outcomes [43,44]. The results, shown in Table 4, highlight the significant influence of the CDH and CLAHE layers on the CCFM-CapsNet model.

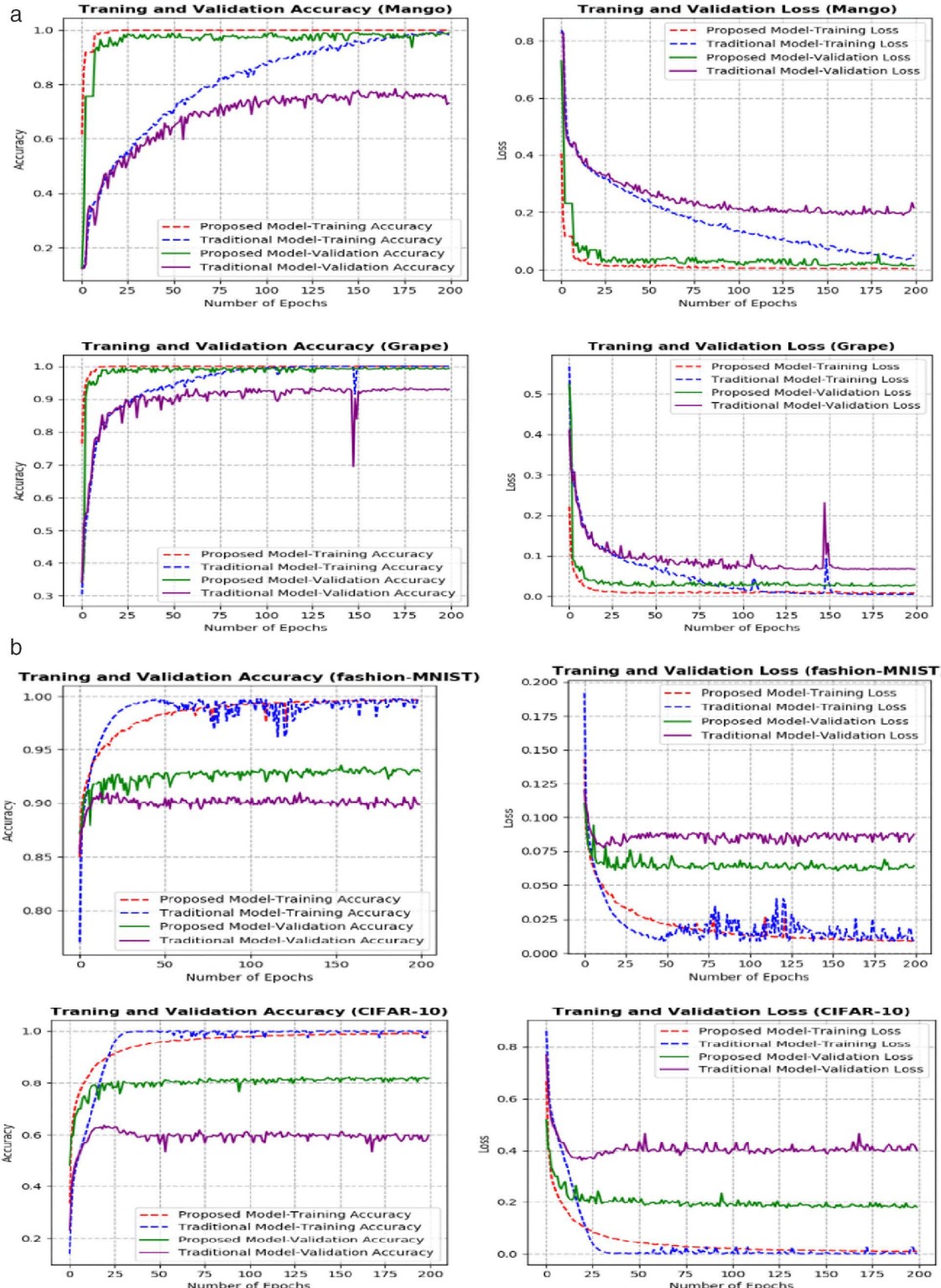

**Fig 2. a**: Accuracy and Loss graphs for (mango and grape datasets) the CCFM-CapsNet model and original/traditional CapsNet models. **b**: Accuracy and Loss graphs for (fashion-MNIST, and CIFAR-10 datasets) the CCFM-CapsNet model and original/ traditional CapsNet models.

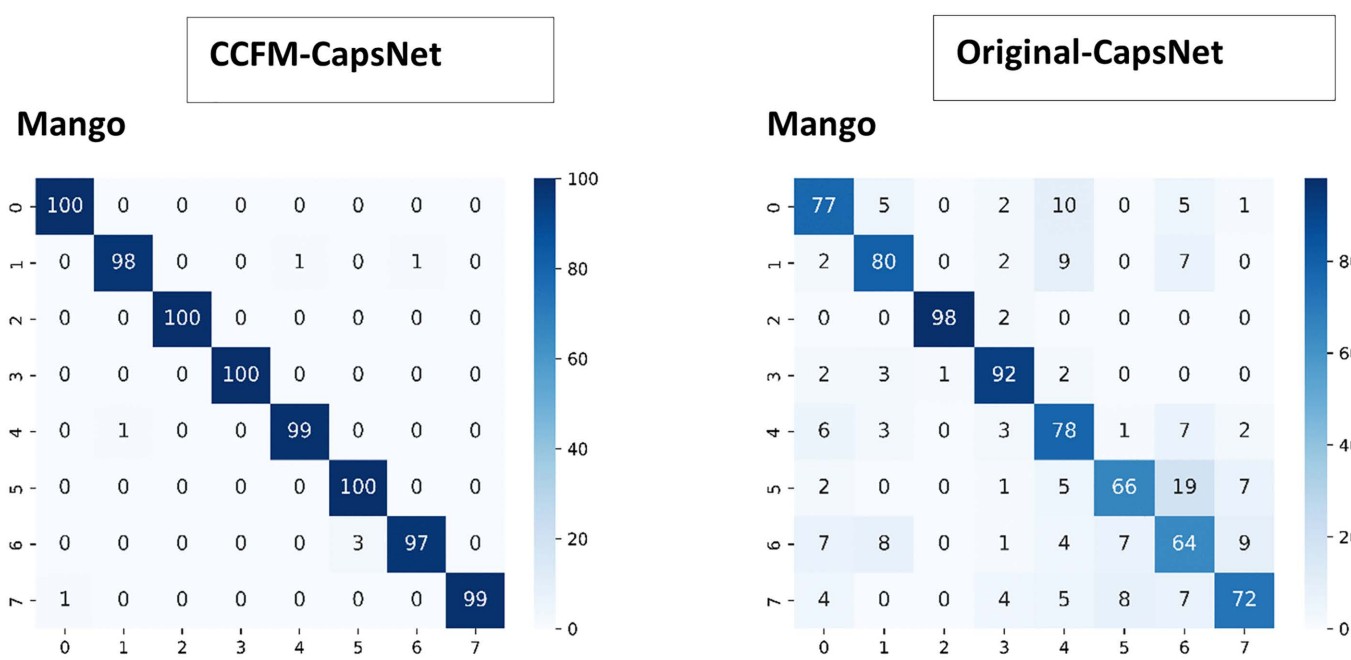

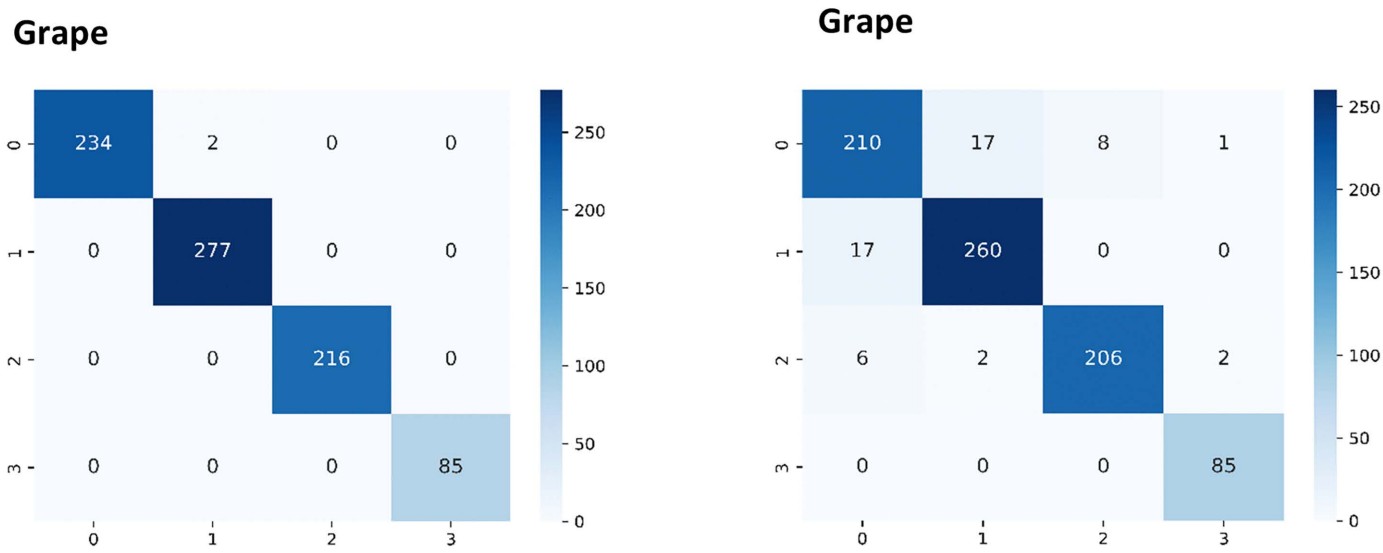

**Fig 3. Confusion matrices for both traditional and CCFM-CapsNets using the datasets for Mangoes and Grapes.**

### 4.3. Number of parameters and size on disk

Deep learning models often create a huge number of parameters, which slows down computations and takes up a lot of memory. This makes them difficult to use on devices with limited resources. However, the CCFM-CapsNet model

**Table 2. Performance metrics for both traditional and CCFM-CapsNets using the dataset of mango.**

| Models (Dataset) | Classes | FP | TP | TN | FN | ACCU (%) | PRE (%) | SENS (%) | SPEC (%) | F1 (%) | Data Size |
|---|---|---|---|---|---|---|---|---|---|---|---|
| Original/Traditional-CapsNet (Mango) | 0 | 23 | 77 | 677 | 23 | 94.25 | 77.00 | 77.00 | 96.71 | 77.00 | 100 |
| | 1 | 19 | 80 | 681 | 20 | 95.13 | 80.81 | 80.00 | 97.29 | 80.40 | 100 |
| | 2 | 1 | 98 | 699 | 2 | 99.63 | 98.99 | 98.00 | 99.86 | 98.49 | 100 |
| | 3 | 15 | 92 | 685 | 8 | 97.13 | 85.98 | 92.00 | 97.86 | 88.89 | 100 |
| | 4 | 35 | 78 | 665 | 22 | 92.88 | 69.03 | 78.00 | 95.00 | 73.24 | 100 |
| | 5 | 16 | 66 | 684 | 34 | 93.75 | 80.49 | 66.00 | 97.71 | 72.53 | 100 |
| | 6 | 45 | 64 | 655 | 36 | 89.88 | 58.72 | 64.00 | 93.57 | 61.25 | 100 |
| | 7 | 19 | 72 | 681 | 28 | 94.13 | 79.12 | 72.00 | 97.29 | 75.39 | 100 |
| CCFM-CapsNet (Mango) | 0 | 1 | 100 | 699 | 0 | 99.88 | 99.01 | 100 | 99.86 | 99.50 | 100 |
| | 1 | 1 | 98 | 699 | 2 | 99.63 | 98.99 | 98.00 | 99.86 | 98.49 | 100 |
| | 2 | 0 | 100 | 700 | 0 | 100 | 100 | 100 | 100 | 100 | 100 |
| | 3 | 0 | 100 | 700 | 0 | 100 | 100 | 100 | 100 | 100 | 100 |
| | 4 | 1 | 99 | 699 | 1 | 99.75 | 99.00 | 99.00 | 99.86 | 99.00 | 100 |
| | 5 | 3 | 100 | 697 | 0 | 99.63 | 97.09 | 100 | 99.57 | 98.52 | 100 |
| | 6 | 1 | 97 | 699 | 3 | 99.50 | 98.98 | 97.00 | 99.86 | 97.98 | 100 |
| | 7 | 0 | 99 | 700 | 1 | 99.88 | 100 | 99.00 | 100 | 99.50 | 100 |

**Table 3. Performance metrics for both traditional and CCFM-CapsNets on the dataset of grape.**

| Models (Dataset) | Classes | FP | TP | TN | FN | ACCU (%) | PRE (%) | SENS (%) | SPEC (%) | F1 (%) | Data Size |
|---|---|---|---|---|---|---|---|---|---|---|---|
| Original/Traditional-CapsNet (Grape) | 0 | 23 | 210 | 553 | 26 | 93.97 | 90.13 | 88.98 | 96.01 | 89.55 | 236 |
| | 1 | 19 | 260 | 516 | 17 | 95.57 | 93.19 | 93.86 | 96.45 | 93.52 | 277 |
| | 2 | 8 | 206 | 588 | 10 | 97.78 | 96.26 | 95.37 | 98.66 | 95.81 | 216 |
| | 3 | 3 | 85 | 724 | 0 | 99.63 | 96.59 | 100 | 99.59 | 98.27 | 85 |
| CCFM-CapsNet (Grape) | 0 | 0 | 234 | 576 | 2 | 99.75 | 100 | 99.15 | 100 | 99.57 | 236 |
| | 1 | 2 | 277 | 533 | 0 | 99.75 | 99.28 | 100 | 99.63 | 99.64 | 277 |
| | 2 | 0 | 216 | 596 | 0 | 100 | 100 | 100 | 100 | 100 | 216 |
| | 3 | 0 | 85 | 727 | 0 | 100 | 100 | 100 | 100 | 100 | 85 |

produces significantly fewer parameters. It also outperforms the standard CapsNet and other leading models, meaning it's simpler and trains or makes predictions faster.

The CLAHE and CDH layers boosted performance considerably without adding any learnable parameters to the model. Furthermore, the CCFM-CapsNet compact size means it requires less memory compared to many leading models. This reduction in size is a crucial step in making capsule networks practical for hardware-constrained devices, such as IoT gadgets and mobile phones. For a detailed comparison of the parameter counts for the CCFM-CapsNet and traditional CapsNet across different datasets, please refer to Table 5.

### 4.4. Prediction and reconstruction

The initial columns of Fig 8 showcase the class labels and the predicted images that the class capsules then reconstructed, with the complete reconstructions also displayed in the figure. A strange issue was observed with CIFAR-10: the reconstructed images came out entirely white. This unusual outcome likely stems from the complex characteristics inherent in the CIFAR-10 dataset itself. In evaluating how well the traditional CapsNet model, it was observed that out of

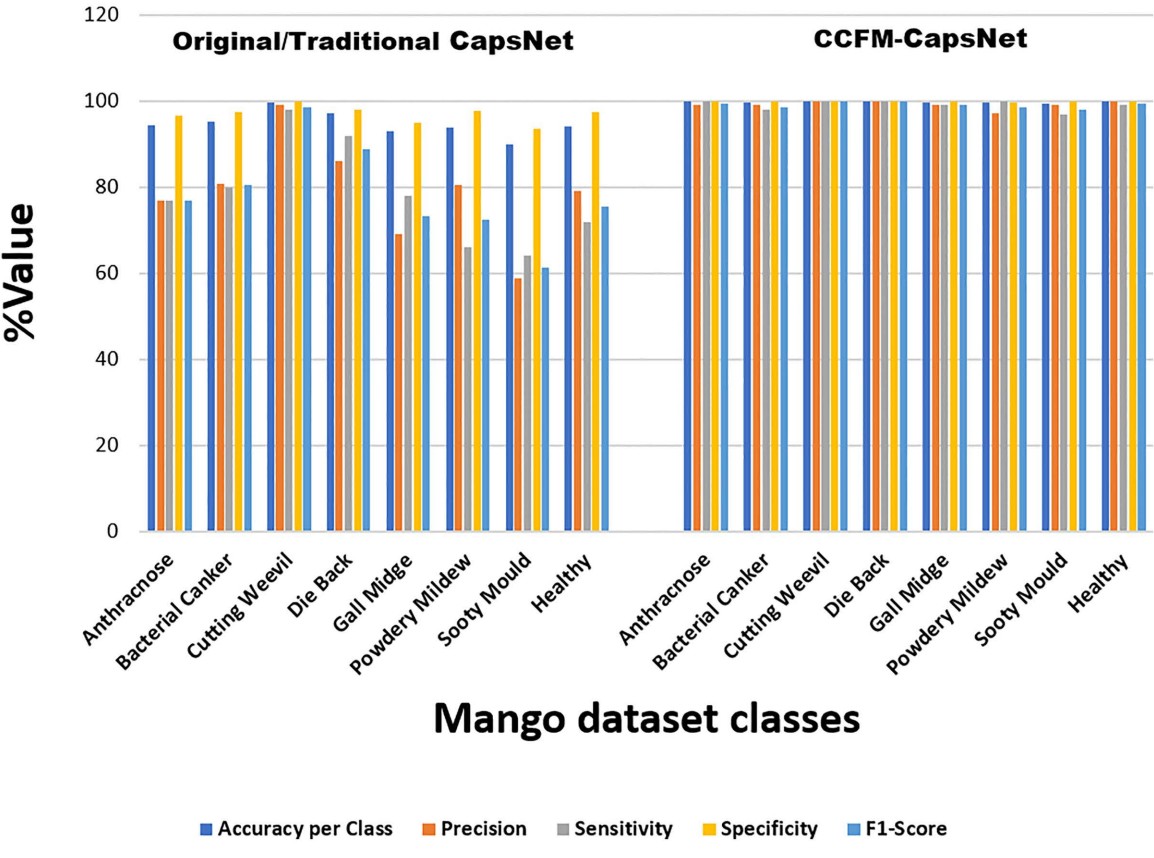

**Performance metrics on the Mango dataset for the Original and the CCFM-CapsNet models**

Original/Traditional CapsNet   CCFM-CapsNet

%Value

Mango dataset classes

■ Accuracy per Class  ■ Precision  ■ Sensitivity  ■ Specificity  ■ F1-Score

**Fig 4. Evaluation outcomes of both traditional and CCFM-CapsNets on the dataset of mango.**

10 predictions, 8 were correctly classified while 2 were misclassified. Notably, one such misclassification involved class 3 being predicted as class 5. Additionally, the prediction confidence for the CapsNet model tended to be relatively low. In contrast, the CCFM-CapsNet model demonstrated superior performance, yielding high-confidence predictions across all instances with no misclassifications recorded. Conversely, the Fashion-MNIST and plant disease datasets produced accurate predictions. Still, the traditional model generated lower prediction scores than the proposed CCFM-CapsNet model, which not only delivered higher scores but also retained finer details and produced sharper images. Fig 8 shows the class labels and reconstructed images using the mango and CIFAR-10 datasets.

### 4.5. Visualizing activation maps and clusters

DL models are frequently referred to as 'black boxes' because of their intricate internal mechanisms. To enhance the understanding and interpretation of neural networks [43,45,46], activation maps from the CCFM-CapsNet model were visualized and assessed alongside original CapsNet visualizations for analytical purposes. Fig 9 displays the activation maps from the initial convolution layer and the PC layer of the CCFM-CapsNet for Lane2, assessed alongside original CapsNet maps from the convolution and PC layers, using the Apple dataset. By examining these activation maps, it is

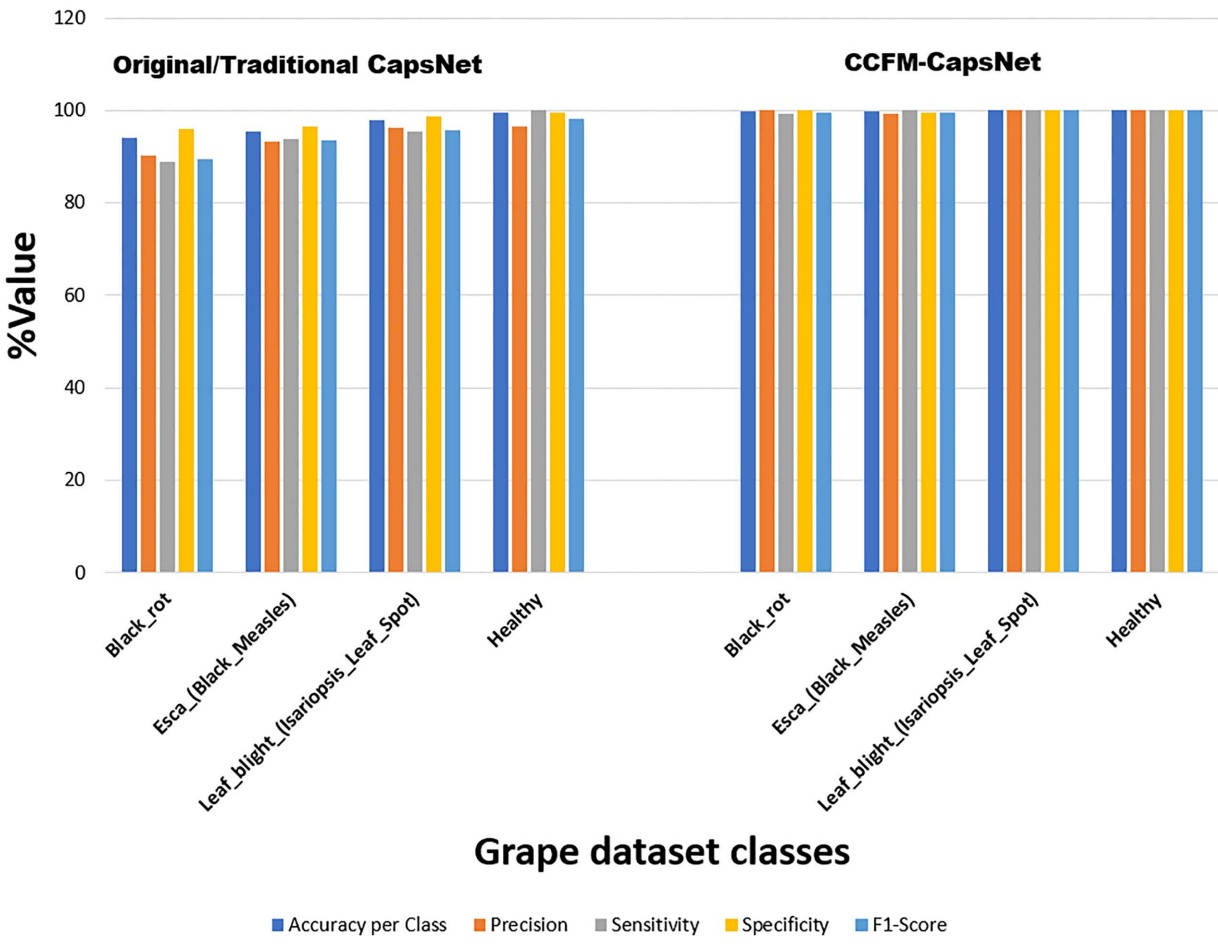

**Fig 5. Evaluation outcomes of CCFM-CapsNet against the original CapsNet on the Grape dataset.**

clear that the CCFM-CapsNet model effectively identifies key features within the input images. On the other hand, the activation maps obtained from the original CapsNet appear distorted, suggesting inferior performance. This trend is consistently observed across all eleven datasets analyzed. Additionally, Fig 10 presents the clustering achieved by the plant_Disease caps layer of the CCFM-CapsNet and the Class capsule layer of a traditional CapsNet on the CIFAR-10, potato, and rice datasets. The overlapping clusters observed in the plant_Disease caps layer shows the capability of the CCFM-CapsNet model in identifying similar characteristics across various classes present in the datasets. It can be observed that the plant_Disease caps layer in the CCFM-CapsNet model groups data from the various datasets more tightly than the traditional CapsNet. In contrast, the traditional CapsNet produces less cohesive clusters, with numerous data points lying significantly outside their main groupings.

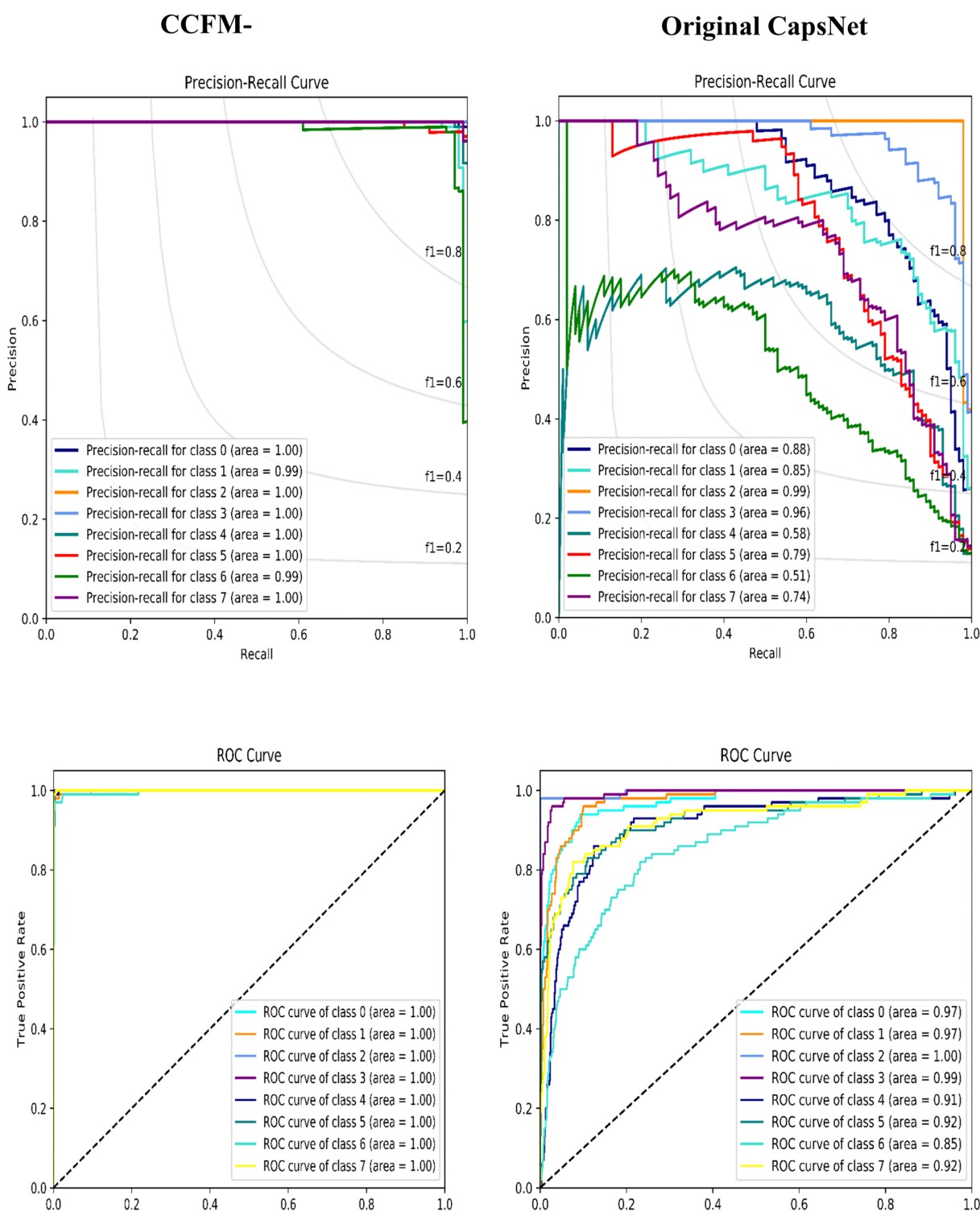

**Fig 6. Evaluation of CCFM-CapsNet and Original CapsNet via PR and ROC curves on the mango dataset.**

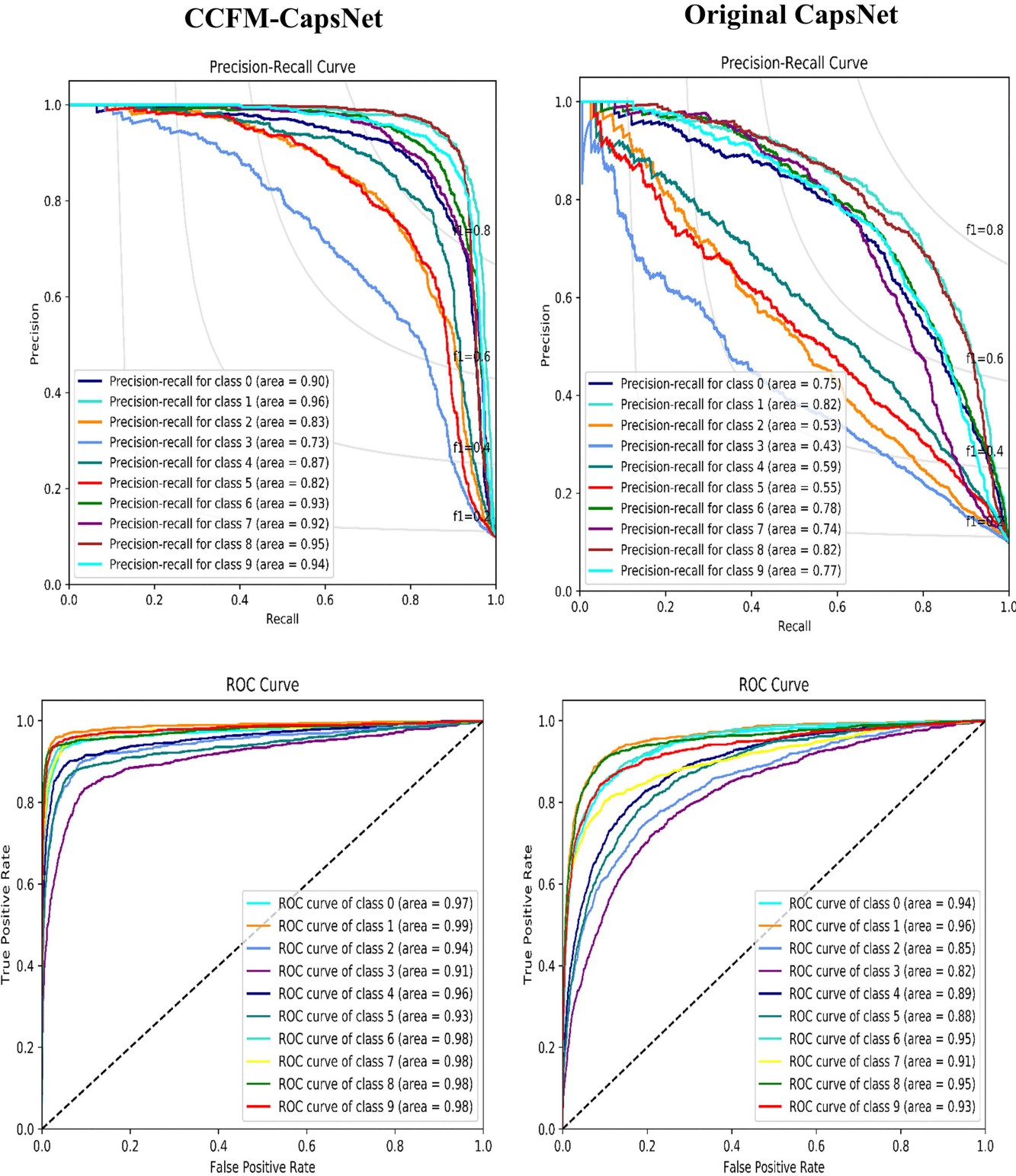

**Fig 7. Evaluation of CCFM-CapsNet and Original CapsNet via PR and ROC curves on the CIFAR-10 dataset.**

**Table 4. Results from the Ablation study.**

| Layers | Validation accuracy (%) | |
|---|---|---|
| | **Mango** | **Grape** |
| -CDH | 82.12 | 90.17 |
| -CLAHE | 97.75 | 95.95 |
| -Conv1 & Conv3 | 98.62 | 98.88 |
| -MP1 & MP3 | 97.88 | 97.17 |
| -Conv2 & Conv4 | 98.75 | 98.28 |
| -MP2 & MP4 | 98.88 | 99.14 |
| -Dropout | 98.13 | 97.17 |
| **+All Layers** | **99.13** | **99.75** |

**Table 5. Generated parameter count (Pc)(Millions (M)) and Size on disk (S)(MB) comparison.**

| CapsNet Models/References | Pc(M) and S(MB) | | | | | | | | | | | |
|---|---|---|---|---|---|---|---|---|---|---|---|---|
| | | **Apple** | **Grape** | **Banana** | **Maize** | **Pepper** | **Mango** | **Potato** | **Tomato** | **Rice** | **Fashion-MNIST** | **CIFAR-10** |
| Dual-Input CapsNet [17] | Pc | * | * | * | * | * | * | * | 6.04 | * | * | 5.48 |
| Shallow/Multi-Input CapsNet [19] | Pc | * | * | * | * | * | * | * | 4.1/ 4.0 | * | 2.50/ 2.20 | 4.60/ 4.30 |
| Gabor-Maxpooled CapsNet [20] | Pc | * | * | * | * | * | * | * | 8.71 | * | * | * |
| CapsNet [23] | Pc | * | * | * | * | * | * | 9.86 | | * | * | * |
| K-Means CapsNet [24] | Pc | * | * | * | * | * | * | * | 5.12 | * | * | * |
| CapsNet [26] | Pc | * | * | * | 8.40 | * | * | * | 8.40 | * | 2.80 | 5.20 |
| Gabor CapsNet [28] | Pc | * | * | * | * | * | * | * | 12.00 | * | * | * |
| Traditional CapsNet [11] | Pc | 10.13 | 10.13 | 10.13 | 10.13 | 9.59 | 11.21 | 9.86 | 11.75 | 10.13 | 8.22 | 11.75 |
| | S | 38.6 | 38.6 | 38.6 | 38.6 | 36.5 | 42.7 | 37.6 | 44.8 | 38.6 | 31.3 | 44.8 |
| **CCFM-CapsNet Proposed** | Pc | **4.69** | **4.69** | **4.69** | **4.69** | **4.63** | **4.79** | **4.66** | **4.84** | **4.69** | **2.39** | **4.84** |
| | S | **17.9** | **17.9** | **17.9** | **17.9** | **17.7** | **18.3** | **17.8** | **18.5** | **17.9** | **9.16** | **18.5** |

## 4.6. Results comparison

We compared the efficiency of our CCFM-CapsNet method against traditional methods and other leading-edge models. Table 6 summarizes the accuracy achieved by each model. While our improvements focused on the network's architecture through dynamic routing, the comparison includes models with both algorithmic and structural changes, as well as those employing different routing algorithms. Table 6 shows the suggested CCFM-CapsNet model notably outperformed the traditional model across all eleven datasets, achieving validation accuracy improvements of 6.62%, 14.29%, 6.26%, 4.81%, 20.75%, 40.32%, 4.41%, 0.76%, 9.95%, 18.76%, and 2.5% for the datasets of apples, bananas, grapes, maize, mangoes, pepper, rice, tomatoes, CIFAR-10, and Fashion-MNIST datasets, respectively. Additionally, the CCFM-CapsNet demonstrated reductions in the number of parameters by 5.44M, 5.44M, 5.44M, 5.44M, 6.42M, 4.96M, 5.20M, 5.44M, 6.91M, 6.91M, and 5.83M, and disk size reductions of 20.70MB, 20.70MB, 20.70MB, 20.70MB, 24.40MB, 18.80MB, 19.80MB, 20.70MB, 26.30MB, 26.30MB, and 22.14MB, respectively, as opposed to the traditional CapsuleNet. Full details are presented in Table 5. The reduced parameter counts and smaller disk capacity of the CCFM-CapsNet model highlight its fitness for implementation on low-resource and computationally-restricted devices. As shown in Tables 2 and 3, the CCFM-CapsNet outperformed the original CapsNet method in the context of overall sensitivity, precision, F1-score and specificity. Specifically, it

**Mango – CCFM-CapsNet**

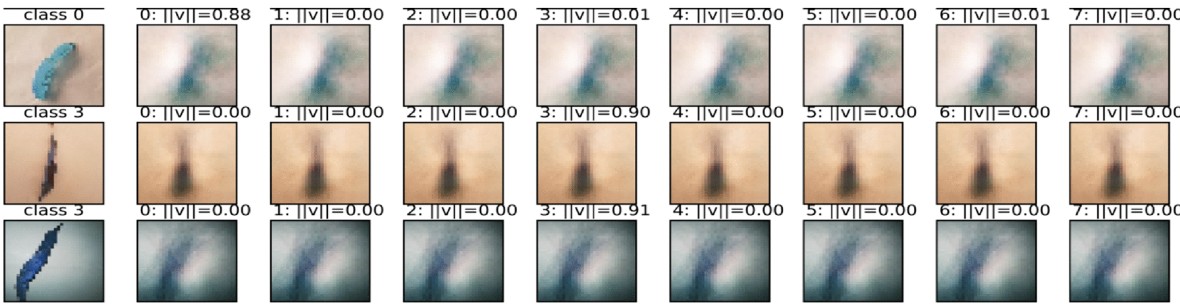

**Mango –Original-CapsNet**

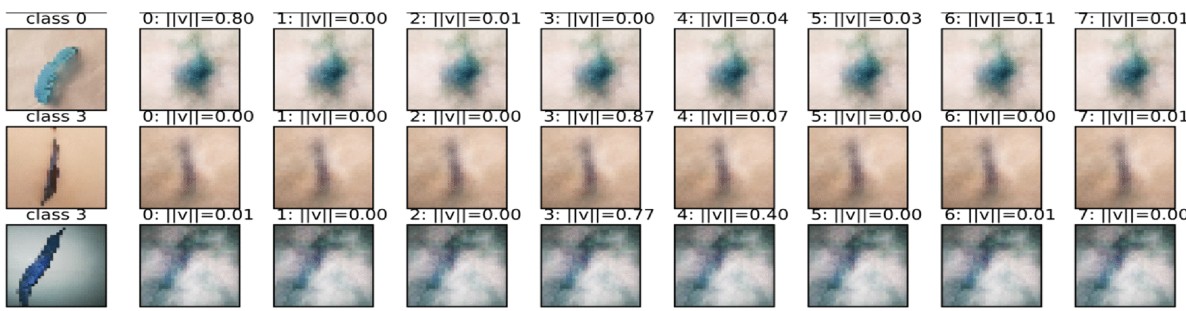

**CIFAR-10 –CCFM-CapsNet**

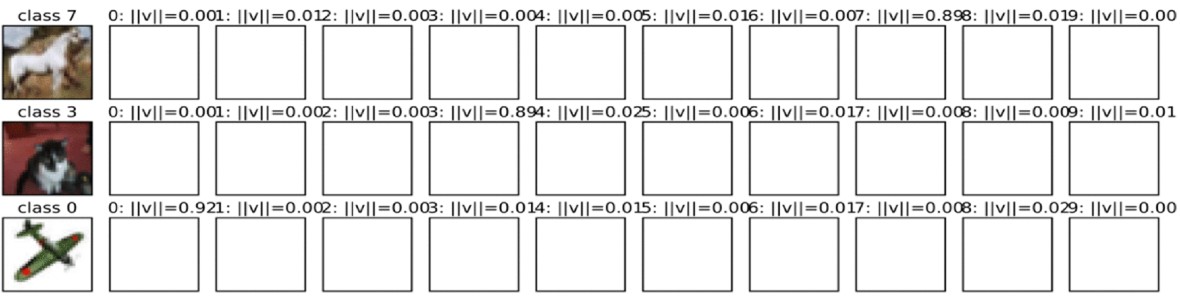

**CIFAR-10 –Original-CapsNet**

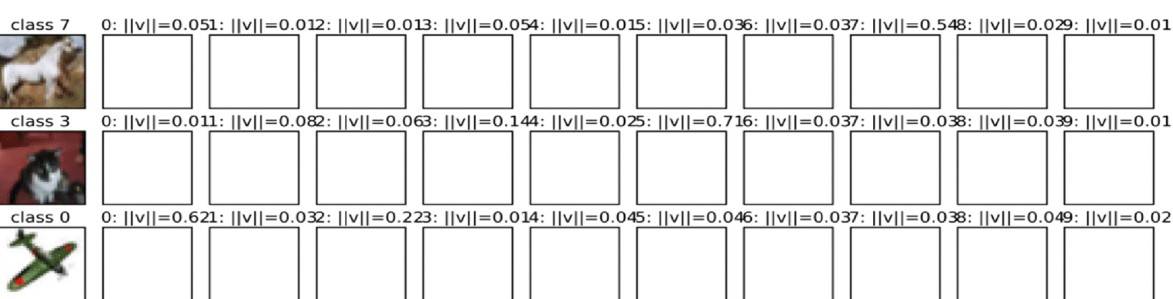

**Fig 8. Images reconstructed by CCFM-CapsNet and Original CapsNet using the mango and CIFAR-10 datasets.**

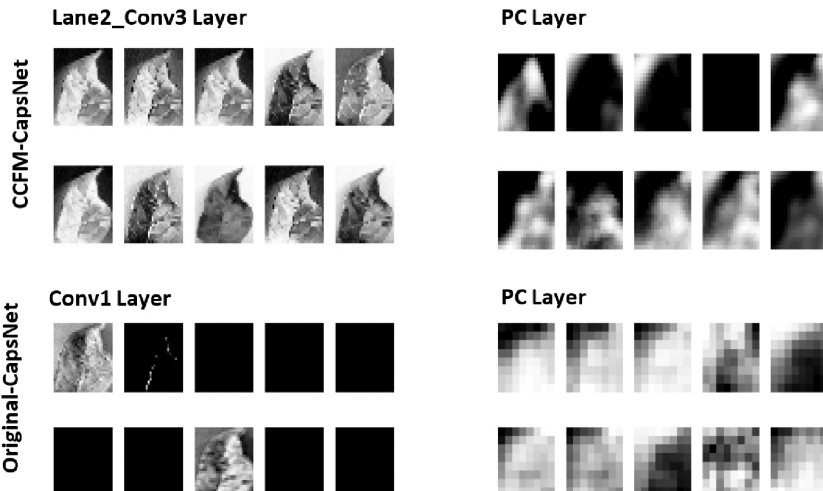

**Fig 9. Feature map of both the traditional and CCFM-CapsNets after training on the dataset of Apple.**

achieved improvements of 20.36%, 20.75%, 2.97%, and 20.72% on the Mango dataset, and 5.78%, 5.24%, 2.23%, and 5.51% on the Grape dataset, respectively, across the same metrics. Furthermore, based on the Mango dataset, the CCFM-CapsNet architecture surpassed the original CapsNet with respect to PR and ROC outcomes by 5.87% and 21.00%, respectively, as shown in Fig 6. Similarly, using the CIFAR-10 dataset, it achieved improvements of 5.4% in ROC and 20.70% in PR, as illustrated in Fig 7. Moreover, Figs 8–10 show that the predictions, and reconstructions activation maps, and class clusters produced by the proposed CCFM-CapsNet were superior to those of the original CapsNet contributing to XAI.

Additionally, the CCFM-CapsNet demonstrated performance that was matching leading architectures presented in prior studies. This exceptional performance can be attributed to the effectiveness of the added layers, such as CLAHE, CDH, the three additional convolutional layers, and max-pooling, which boost the encoder network's potential to retrieve important properties.

## 5. Conclusion and future works

This article introduces a novel methodology called CCFM-CapsNet for the classification and detection of plant diseases, and also for classifying CIFAR-10 and fashion-MNIST datasets. Thus, the suggested model CCFM-CapsNet incorporates CLAHE operating as a foundational tier which improves a source image prior to being passed to the high-level convolutional layer and uses CDH to extract significant features. Results from eleven different datasets show that our approach achieves better result compared to results of the baseline traditional Capsule Network and other cutting-edge models. Also results of the proposed CCFM-CapsNet model demonstrate that it can generalise well to unseen images, operates efficiently, and has a low parameter count, reducing computational overhead. Thus, this level of efficiency allows it for implementation on devices with constrained resources. Consequently, our method possesses the ability to serve as a valuable utility for farmers and agricultural experts in detecting plant diseases and also in achieving SDG 2 (i.e., Sustainable Development Goal 2 (i.e., Zero Hunger)), which seeks to end global hunger by the year 2030. Moving forward, we plan to enhance our CCFM-CapsNet's routing mechanism to achieve greater network optimization.

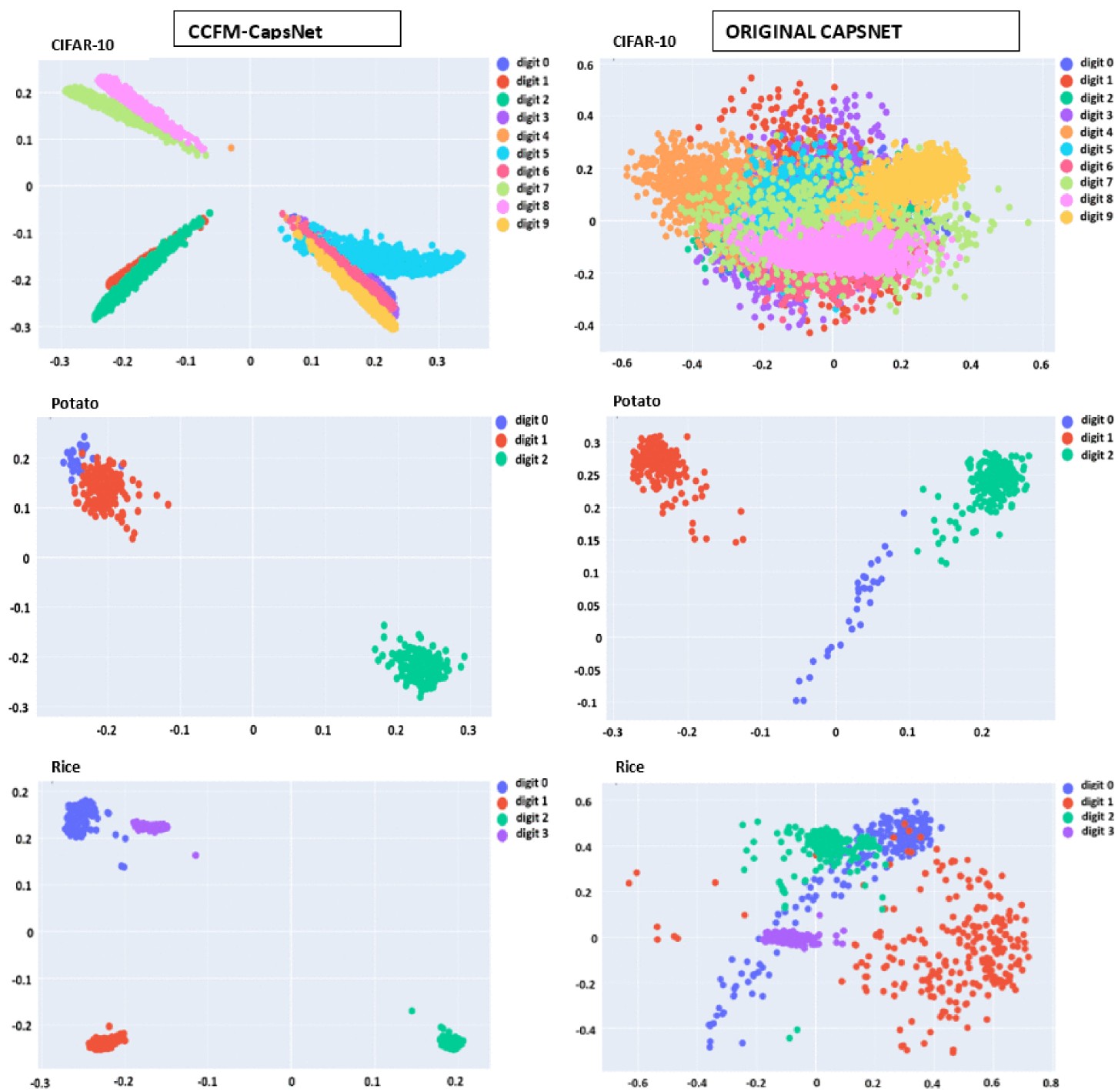

**Fig 10. Clusters formed in the class capsule layers of both the traditional and CCFM-CapsNets using the datasets of CIFAR-10, potato, and rice.**

Table 6. Comparison of previous works and the suggested CCFM-CapsNet model.

| CapsNet Models/References | Validation Accuracy (%) | | | | | | | | | | |
|---|---|---|---|---|---|---|---|---|---|---|---|
| | Apple | Grape | Banana | Corn | Pepper | Mango | Potato | Tomato | Rice | Fashion-MNIST | CIFAR-10 |
| SE-SK CapsNet [15] | * | * | * | * | * | * | * | * | 97.19 | * | * |
| CapsNet [16] | * | * | * | * | * | * | * | * | 96.39 | * | * |
| Dual-Input CapsNet [17] | * | * | * | * | * | * | * | * | 93.03 | 76.58 | * |
| Multi-Channel CapsNet [18] | * | * | * | * | * | * | * | * | 98.15 | * | * |
| Shallow/Multi-Input CapsNet [19] | * | * | * | * | * | * | * | * | 97.33/ 94.04 | 75.75/ 63.95 | 92.70/ 91.45 |
| Gabor-Maxpooled CapsNet [20] | * | * | * | * | * | * | * | * | 97.98 | * | * |
| CapsNet [21] | * | 95.00 | * | * | * | * | * | * | * | * | * |
| CapsNet [22] | * | * | * | * | 95.76 | * | * | * | * | * | * |
| CapsNet [23] | * | * | * | * | * | * | 91.83 | * | * | * | * |
| K-Means CapsNet [24] | * | * | * | 97.99 | * | * | * | * | 98.80 | * | * |
| Dilated CapsNet [25] | 93.16 | * | * | * | * | * | * | * | * | * | * |
| CapsNet [26] | * | * | * | 96.79 | * | * | * | * | 98.06 | 75.80 | 92.72 |
| E-GAN CapsNet [27] | * | * | 97.63 | * | * | * | * | * | * | * | * |
| Gabor CapsNet [28] | * | * | * | * | * | * | * | * | 98.13 | * | * |
| ConvCapsNet [30] | * | * | 99.12 | * | * | * | * | * | * | * | * |
| Original CapsNet [11] | 92.91 | 93.49 | 80.95 | 92.59 | 59.68 | 78.38 | 95.36 | 88.59 | 99.24 | 90.98 | 63.58 |
| **CCFM-CapsNet Proposed** | **99.53** | **99.75** | **95.24** | **97.40** | **100** | **99.13** | **99.77** | **98.54** | **100** | **93.48** | **82.34** |

## Author contributions

**Conceptualization:** Steve Okyere-Gyamfi.

**Data curation:** Steve Okyere-Gyamfi, Vivian Akoto-Adjepong.

**Methodology:** Steve Okyere-Gyamfi.

**Project administration:** Michael Asante, Kwame Ofosuhene Peasah, Yaw Marfo Missah.

**Resources:** Vivian Akoto-Adjepong.

**Supervision:** Michael Asante, Kwame Ofosuhene Peasah, Yaw Marfo Missah.

**Validation:** Michael Asante, Kwame Ofosuhene Peasah, Yaw Marfo Missah, Vivian Akoto-Adjepong.

**Visualization:** Steve Okyere-Gyamfi.

**Writing – original draft:** Steve Okyere-Gyamfi.

**Writing – review & editing:** Michael Asante, Kwame Ofosuhene Peasah, Yaw Marfo Missah, Vivian Akoto-Adjepong.

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
