## [Decision Letter · Decision Letter 0]

15 Jul 2025

PONE-D-25-27327
Contrast Limited Adaptive Histogram Equalization (CLAHE) and Colour Difference Histogram (CDH) Feature Merging Capsule Network (CCFMCapsNet) for Complex image recognition
PLOS ONE

Dear Dr Okyere-Gyamfi,

Thank you for submitting your manuscript to PLOS ONE. After careful consideration, we feel that it has merit but does not fully meet PLOS ONE’s publication criteria as it currently stands. Therefore, we invite you to submit a revised version of the manuscript that addresses the points raised during the review process.

We look forward to receiving your revised manuscript.

Kind regards,

Sunil Kumar Sharma

Academic Editor

PLOS ONE

Journal Requirements:

Reviewers' comments:

Reviewer's Responses to Questions

**Comments to the Author**

1. Is the manuscript technically sound, and do the data support the conclusions?

Reviewer #1: Yes

Reviewer #2: Yes

2. Has the statistical analysis been performed appropriately and rigorously? 

Reviewer #1: Yes

Reviewer #2: N/A

3. Have the authors made all data underlying the findings in their manuscript fully available?

Reviewer #1: Yes

Reviewer #2: Yes

4. Is the manuscript presented in an intelligible fashion and written in standard English?

Reviewer #1: Yes

Reviewer #2: Yes

5. Review Comments to the Author

Reviewer #1: This manuscript introduces CCFM-CapsNet, an enhanced Capsule Network model integrating CLAHE and CDH to improve image classification in plant disease detection and standard benchmarks. While the proposed architecture shows promising results, the manuscript has several issues. Substantial revisions are required for the manuscript.

1. In Section 3.5 (Datasets), the class label numbering for each dataset (e.g., “3,2,1,0” for apple, grape, corn) is inconsistent and sometimes confusing. The order of class labels changes between datasets, and explanations of which number corresponds to which class are not always clear. I recommend unifying the labeling convention across datasets and presenting the mappings between class numbers and disease categories in a clear table or standardized list.

2. While the manuscript provides many architectural details, the training procedures lack clarity in important aspects: there is no hyperparameter tuning processes, or evaluation protocols for train/validation/test splits beyond the 80:20 mention. I recommend providing a more thorough, step-by-step description of the experimental pipeline.

3. The manuscript compares CCFM-CapsNet primarily with traditional and modified CapsNet models but does not include comparisons with recent state-of-the-art non-CapsNet architectures (e.g., ViT, EfficientNet). Including such benchmarks or at least a discussion of their relevance would significantly strengthen your claims about the proposed model’s performance.

Reviewer #2: 1. What measures you took to ensure that model is not overfitted?

2. Perform k-fold validation, with minimum 10 Folds.

3. Improve image quality, few images are very blur and hard to understand.

4. DOI: 10.1134/S1054661825700087, also claimed more than 99% accuracy on Plant-Village dataset. This is the same dataset which you used in the article. How your research is more promising then the above mentioned.

6. PLOS authors have the option to publish the peer review history of their article (what does this mean?). If published, this will include your full peer review and any attached files.

Reviewer #1: No

Reviewer #2: No

---

## [Author Response · Author response to Decision Letter 1]

28 Aug 2025

Original Manuscript ID: PONE-D-25-27327

Original Article Title: “Contrast Limited Adaptive Histogram Equalization (CLAHE) and Colour Difference Histogram (CDH) Feature Merging Capsule Network (CCFMCapsNet) for Complex image recognition”

To: PLOS ONE

Re: Response to Reviewers

Dear Editor,

We would like to express our profound gratitude to the Editor and the reviewers for their professional and constructive comments on our manuscript, “Contrast Limited Adaptive Histogram Equalization (CLAHE) and Colour Difference Histogram (CDH) Feature Merging Capsule Network (CCFMCapsNet) for Complex image recognition” (ID: PONE-D-25-27327). We have revised the manuscript according to the comments and suggestions of the academic Editor and the reviewers, and have responded to the academic editor's and the reviewers’ concerns. The responses to the reviewer’s comments are therefore attached.

Thank you for allowing a resubmission of our manuscript, with an opportunity to address the academic editor and the reviewers’ comments.

We are uploading (a) our rebuttal letter that responds to each of the comment (below) (Response to Reviewers), (b) an updated marked-up manuscript with yellow and green highlighting indicating changes made as suggested by Reviewer #1 and Reviewer #2 respectively ('Revised Manuscript with Track Changes), and (c) an unmarked, updated manuscript without highlights (Manuscript). Finally, the authors would like to affirm your office that the manuscript can be assigned to the same reviewers. For further information, please feel free to contact the corresponding author. His personal information is as follows:

Mailing address: Mr. Steve Okyere-Gyamfi

Department of Computing and Information Sciences,

Catholic University of Ghana,

P. O. Box 363, Fiapre - Sunyani, Ghana

Phone number: +233 (0) 263299901

Email: steve.og@cug.edu.gh

Best regards

Steve Okyere-Gyamfi et al.

Reviewer #1: This manuscript introduces CCFM-CapsNet, an enhanced Capsule Network model integrating CLAHE and CDH to improve image classification in plant disease detection and standard benchmarks. While the proposed architecture shows promising results, the manuscript has several issues. Substantial revisions are required for the manuscript.

Reviewer #1, Comment # 1: In Section 3.5 (Datasets), the class label numbering for each dataset (e.g., “3,2,1,0” for apple, grape, corn) is inconsistent and sometimes confusing. The order of class labels changes between datasets, and explanations of which number corresponds to which class are not always clear. I recommend unifying the labeling convention across datasets and presenting the mappings between class numbers and disease categories in a clear table or standardized list.

Response: Thank you for pointing out the inconsistency. We have revised Section 3.5 to standardize the labeling format across all datasets. Table I has been added, clearly mapping each dataset’s class labels to their corresponding disease or categories. This enhances readability and clarity throughout the revised manuscript. [see page: 13-15]

Author action:

3.5 Datasets

Eleven independent datasets (i.e., nine plant disease datasets from Kaggle, Mendley, etc, and two benchmark datasets) were used independently to run and assess the original and CCFM-CapsNets effectiveness, and compared with various cutting-edge CapsNet models found in the literature that used the same datasets to evaluate their performance. This includes the Apple, Corn, Grape, Pepper, Tomato, Potato, Banana, Mango, Rice, Fashion-MNIST, and CIFAR-10 datasets.

The datasets of Corn, Apple, Pepper, Tomato, Grape, and Potato were sized 256x256, and are included in the Plant Village dataset (34).

Banana: Consists of varying dimensions ranging from 2230x4000 to 3120x4208 (35).

Mango: Consists of varying dimensions ranging from 240x240 to 320x240 (36).

Rice: Consists of varying dimensions ranging from 216x289 to 344x516 (37).

Fashion-MNIST: Consists of grayscale images, of size 28x28, and is complex (38).

CIFAR-10: consists of 32x32x3. They have different backgrounds and are more complex than Fashion-MNIST (39).

Table I presents the total image count for each of the 11 datasets, along with the number of classes and the sample size per class.

Most of the datasets for plant disease are significantly imbalanced, with images that are very similar to each other and backgrounds that are not uniform. The pre-processing step utilized to these datasets was rescaling the images to a dimension of 32 x 32 x 3 to maintain manageability, except for the Fashion-MNIST dataset. Additionally, an 80:20 split technique was employed.

Table I: Summary of the 11 independent datasets used separately for model training and evaluation

S/N Dataset Total no. of images No. of Classes Classes and no. of samples Number of samples (images)

1 Apple 3,171 4 0: Apple_scab 630

 1: Black_rot 621

 2: Cedar_apple_rust 275

 3: healthy 1645

2 Grape 4,062 4 0: Black_rot 1180

 1: Esca_(Black_Measles) 1383

 2: Leaf_blight_(Isariopsis_Leaf_Spot) 1076

 3: healthy 423

3 Corn 3,852 4 0: Cercospora_leaf_spot, Gray_leaf_spot 513

 1: Common_rust 1192

 2: Northern_Leaf_Blight 985

 3: healthy 1162

4 Pepper 2,475 2 0: healthy 1478

 1: Bacterial_spot 997

5 Potato 2,152 3 0: Healthy 152

 1: Early_blight 1000

 2: Late_blight 1000

6 Tomato 18,160 10 0: Bacterial_spot 2127

 1: Early_blight 1000

 2: Late_blight 1909

 3: Leaf_Mold 952

 4: Septoria_leaf_spot 1771

 5: Spider_mites Two-spotted_spider_mite 1676

 6: Target_Spot 1404

 7: Tomato_mosaic_virus 373

 8: Tomato_Yellow_Leaf_Curl_Virus 5357

 9: Healthy 1591

7 Banana 937 4 0: Cordana 162

 1: Pestalotiopsis 173

 2: Sigatoka 473

 3: Healthy 129

8 Mango 4,000 8 0: Anthracnose 500

 1: Bacterial Canker 500

 2: Cutting Weevil 500

 3: Die Back 500

 4: Gall Midge 500

 5: Powdery Mildew 500

 6: Powdery Mildew 500

 7: Healthy 500

9 Rice 5,932 4 0: Bacterial_blight 1584

 1: Blast 1440

 2: Brown_spot 1600

 3: Tungro 1308

10 Fashion-MNIST 70,000 10 0: T-shirt/top 7000

 1: Trouser 7000

 2: Pullover 7000

 3: Dress 7000

 4: Coat 7000

 5: Sandal 7000

 6: Shirt 7000

 7: Sneaker 7000

 8: Bag 7000

 9: Ankle boot 7000

11 CIFAR-10 60,000 10 0: airplane 6000

 1: automobile 6000

 2: bird 6000

 3: cat 6000

 4: deer 6000

 5: dog 6000

 6: frog 6000

 7: horse 6000

 8: ship 6000

 9: truck 6000

Reviewer #1, Comment # 2: While the manuscript provides many architectural details, the training procedures lack clarity in important aspects: there is no hyperparameter tuning processes, or evaluation protocols for train/validation/test splits beyond the 80:20 mention. I recommend providing a more thorough, step-by-step description of the experimental pipeline.

Response: We thank the reviewer for the valuable feedback regarding the clarity of our training procedures. In response, we have expanded the description of our experimental pipeline in the revised manuscript to provide a more thorough and step-by-step explanation of the hyperparameter tuning processes, the experimental pipeline, and the evaluation metrics used. [see page: 15-18]

Author action:

 Experimental setup

3.6.1 Hyperparameter Tuning:

The following hyperparameters were chosen for performance evaluation of both the original and CCFM-CapsNetss training and testing, and compared with various cutting-edge CapsNet models found in the literature that used the same datasets to assess their performance:

 Learning rate: 0.001 with exponential decay (decay rate = 0.9)

 Batch size: 100

 Epochs: 200

 Routing iterations: 3

 Reconstruction loss coefficient (λ): 0.392

 All images in the various datasets were resized to 32X32X3, except Fashion-MNIST, which was 28x28x1

 For each of the 11 independent datasets used in this study, we applied an 80:20 train-test split. All splits were stratified to maintain class distribution.

[see page: 15-16]

3.6.2 Experimental Pipeline:

 All model training and testing took place in Keras, running on top of TensorFlow, using a 64-bit Windows computer equipped with an NVIDIA GeForce RTX 2080 SUPER GPU (8 GB).

 Training was performed separately for each of the 11 datasets using the same pipeline to ensure fair comparisons.

 The top-performing model was saved and used for evaluation.

 The original CapsNet implementation from Xifeng Guo from Sabour was used as the base and modified to accommodate the proposed architectural enhancements.

 Evaluation metrics included precision, accuracy, sensitivity or recall, F1-score, specificity, ROC-AUC, and PR-AUC.

[see page: 16]

3.7 Performance Evaluation Measures

The study employed a variety of metrics to assess classification performance:

 Confusion Matrix: Offers a comprehensive summary of both accurate and inaccurate predictions, enabling the computation of metrics such as sensitivity (recall), precision, specificity, F1-Score, and accuracy from TP-true positive, FP-false positive, TN-true negative, FN-false negative.

 Validation Accuracy: indicates the ratio of samples correctly classified compared to the overall number of instances. The overall validation accuracy reported reflects outcomes across all experiments, as shown in equation 11.

Accuracy= (TP+TN)/(TP+FP+TN+FN) [11]

 Loss: Measures how far the model’s predictions deviate from the true labels. Margin loss was specifically used during assessment.

 Precision (P): Measure of how many of the predicted positives are truly positives. This is shown in equation 12.

Precision(P)=TP/(TP+FP) [12]

 Recall (R) / Sensitivity: Measures how many of the actual positive instances are correctly detected by the model. This is shown in equation 13.

Recall(R)/Sensitivity=TP/(TP+FN) [13]

 Specificity: The proportion of true negatives identified among all actual negatives. This is shown in equation 14.

Specificity=TN/(TN+FP) [14]

 F1-Score: Combines precision and recall through their harmonic mean, ensuring a trade-off between them. This is shown in equation 15.

F1-Score=2((P*R)/(P+R)) [15]

 Area Under the Curve (AUC): Performance evaluation of models are assessed with Receiver Operating Characteristic (ROC) and Precision–Recall (PR) curves, which are especially informative for imbalanced datasets. Larger AUC values signify a strong discriminative capability.

 Clustering Analysis: To gain insight into the feature distribution, t-SNE was used to visualize and interpret class capsule clusters generated by the model.

 Model Complexity: The study also measured performance by analyzing computational efficiency, expressed through parameter count and memory usage.

[see page: 16-17]

Reviewer #1, Comment # 3: The manuscript compares CCFM-CapsNet primarily with traditional and modified CapsNet models but does not include comparisons with recent state-of-the-art non-CapsNet architectures (e.g., ViT, EfficientNet). Including such benchmarks or at least a discussion of their relevance would significantly strengthen your claims about the proposed model’s performance.

Response: Thank you for the suggestion. While our study’s main focus is on enhancing CapsNet, we agree that at least a discussion of the relevance of recent state-of-the-art non-CapsNet architectures (e.g., ViT, EfficientNet) would significantly strengthen claims of the proposed model’s performance. Hence a discussion of such recent state-of-the-art non-CapsNet architectures have been discussed in the introduction section of the revised manuscript. [see page: 3-4]

Author action:

Recent advances in image classification (especially in crop disease detection) have been dominated by CNNs such as EfficientNet and ResNet, DenseNet, as well as transformer-based models like the Vision Transformer (ViT) (3)(4)(5)(6)(7)(8). These models have also demonstrated impressive performance across large-scale benchmark datasets such as CIFAR-10, fashion-MNIST, ImageNet, etc., often surpassing human-level accuracy (9). Their success is largely attributable to hierarchical feature extraction (in CNNs) and global self-attention mechanisms (in ViTs), both of which enable strong generalization when large volumes of labeled data are available. For example, EfficientNet employs a joint scaling strategy to proportionally scale the width, depth, and resolution, yielding top-tier accuracy (5). Similarly, ViT leverages patch embeddings and multi-head attention to capture long-range dependencies, pushing the boundaries of image classification performance (6).

While these models are highly influential in advancing the field, they also highlight the gaps that Capsule Networks (CapsNets) are designed to address. CNNs and ViTs often struggle with preserving spatial hierarchies and viewpoint equivariance, which can be critical in fine-grained recognition tasks or cases with limited data, such as those in health, agriculture, etc. (10)(11).

As CNNs go deeper, they need broad data coverage to prevent overfitting (12). They also maintain good performance as the model depth increases; however, this introduces challenges, including significant parameter numbers, amplified complexity, increased memory requirements, and intensive computing requirements. To ensure CNNs and ViTs perform effectively and generalize well to new data, time-consuming and labour-intensive data augmentation techniques are necessary(12).

CapsNets, through their routing-by-agreement mechanism, are inherently designed to encode part–whole relationships, making them more interpretable and potentially more robust in low-data regimes addressing the limitations of CNN’s (11).

Capsules need less training data, are less impacted by class imbalance, and are more resilient to changes in spatial orientation. Despite these benefits of CapsNets, they also have some disadvantages (1). Due to the "crowding" problem, their performance is poor on complex images that have diverse backgrounds(12). This issue affects various datasets such as CIFAR-100, CIFAR-10, wild plant images, medical images, multilabel images, and many more. The Capsule network's sensitivity to the image background makes it prone to misclassification.

Additionally, CapsNet attempts to account for every detail in the image. Due to these properties, the network's performance can decline when processing detailed infected images. The encoder network's ineffective ability to extract features considerably hinders CapsNet's performance (13) (14). Consequently, enhancing the current capsule network algorithm is necessary to effectively classify these images.

To enhance the categorization of plant ailments employing CapsNet, we incorporated CapsNet with dynamic routing (11). Additionally, we introduced CDH in conjunction with CapsNet to extract crucial features for the first time. Moreover, we integrated CLAHE to improve visual clarity by reducing inherent noise, thereby enhancing the feature extraction capabilities of the encoder network.

It is worth highlighting that the current investigation is devoted to the advancement of CapsNet architectures rather than proposing a direct replacement for CNNs or ViTs. The comparisons in this study are therefore made against established CapsNet variants to ensure that the proposed contributions are evaluated fairly within the same model family. Nonetheless, discussing CNN- and transformer-based models remains relevant because they are part of the cutting-edge techniques in image classification. Their strengths and limitations provide a useful backdrop for understanding why improving CapsNets is significant. In particular, CNNs and ViTs, together with their variants are known to require extensive data augmentation and large-scale training datasets to generalize well (6)(7)(8), whereas CapsNets offer a principled approach to capturing spatial hierarchies w

---

## [Decision Letter · Decision Letter 1]

12 Oct 2025

Contrast Limited Adaptive Histogram Equalization (CLAHE) and Colour Difference Histogram (CDH) Feature Merging Capsule Network (CCFMCapsNet) for Complex image recognition

PONE-D-25-27327R1

Dear Dr. Steve Okyere-Gyamfi,

We’re pleased to inform you that your manuscript has been judged scientifically suitable for publication and will be formally accepted for publication once it meets all outstanding technical requirements.

Kind regards,

Giacomo Fiumara, PhD

Academic Editor

PLOS ONE

Additional Editor Comments (optional):

Dear authors,

The second round of reviews of the manuscript entitled "Contrast Limited Adaptive Histogram Equalization (CLAHE) and Colour Difference Histogram (CDH) Feature Merging Capsule Network (CCFMCapsNet) for Complex image recognition" is now complete. The opinion of the Reviewers, and mine, is that the manuscript can be accepted for publication on PLOS ONE.

Best regards,

Giacomo Fiumara

Reviewers' comments:

Reviewer's Responses to Questions

**Comments to the Author**

1. If the authors have adequately addressed your comments raised in a previous round of review and you feel that this manuscript is now acceptable for publication, you may indicate that here to bypass the “Comments to the Author” section, enter your conflict of interest statement in the “Confidential to Editor” section, and submit your "Accept" recommendation.

Reviewer #1: All comments have been addressed

Reviewer #2: (No Response)

2. Is the manuscript technically sound, and do the data support the conclusions?

Reviewer #1: Yes

Reviewer #2: Yes

3. Has the statistical analysis been performed appropriately and rigorously? 

Reviewer #1: Yes

Reviewer #2: N/A

4. Have the authors made all data underlying the findings in their manuscript fully available?

Reviewer #1: Yes

Reviewer #2: Yes

5. Is the manuscript presented in an intelligible fashion and written in standard English?

Reviewer #1: Yes

Reviewer #2: Yes

6. Review Comments to the Author

Reviewer #1: All comments have been addressed. Based on the comprehensive revisions and the satisfactory resolution of all outstanding concerns, I recommend acceptance in its current form.

Reviewer #2: The authors, of the article "Contrast Limited Adaptive Histogram Equalization (CLAHE) and Colour Difference Histogram (CDH) Feature Merging Capsule Network (CCFMCapsNet) for Complex image recognition", have fully answered to all the concerned raised by the reviewers.

7. PLOS authors have the option to publish the peer review history of their article (what does this mean?). If published, this will include your full peer review and any attached files.

Reviewer #1: No

Reviewer #2: No

---

## [Editor Report · Acceptance letter]

PONE-D-25-27327R1

PLOS ONE

Dear Dr. Okyere-Gyamfi,

I'm pleased to inform you that your manuscript has been deemed suitable for publication in PLOS ONE. Congratulations! Your manuscript is now being handed over to our production team.

Kind regards,

on behalf of

Dr. Giacomo Fiumara

Academic Editor

PLOS ONE